# Ratiometric population sensing by a pump-probe signaling system in *Bacillus subtilis*

Heiko Babel [1,2,3], Pablo Naranjo-Meneses[1,2,3], Stephanie Trauth[1,2,3], Sonja Schulmeister[1,2], Gabriele Malengo[3,4], Victor Sourjik [3,4] & Ilka B. Bischofs [1,2,3✉]

Communication by means of diffusible signaling molecules facilitates higher-level organization of cellular populations. Gram-positive bacteria frequently use signaling peptides, which are either detected at the cell surface or 'probed' by intracellular receptors after being pumped into the cytoplasm. While the former type is used to monitor cell density, the functions of pump-probe networks are less clear. Here we show that pump-probe networks can, in principle, perform different tasks and mediate quorum-sensing, chronometric and ratiometric control. We characterize the properties of the prototypical PhrA-RapA system in *Bacillus subtilis* using FRET. We find that changes in extracellular PhrA concentrations are tracked rather poorly; instead, cells accumulate and strongly amplify the signal in a dose-dependent manner. This suggests that the PhrA-RapA system, and others like it, have evolved to sense changes in the composition of heterogeneous populations and infer the fraction of signal-producing cells in a mixed population to coordinate cellular behaviors.

[1] BioQuant Center of the University of Heidelberg, Im Neuenheimer Feld 267, 69120 Heidelberg, Germany. [2] Center for Molecular Biology (ZMBH), University of Heidelberg, Im Neuenheimer Feld 282, 69120 Heidelberg, Germany. [3] Max-Planck-Institute for Terrestrial Microbiology, Karl-von-Frisch Str. 10, 35043 Marburg, Germany. [4] LOEWE Center for Synthetic Microbiology (SYNMIKRO), Karl-von-Frisch Str. 16, 35043 Marburg, Germany. ✉email: ilka.bischofs@mpi-marburg.mpg.de

Cell-to-cell communication by diffusible signaling molecules is a central component of higher-level organization of populations in time and space. Specific signaling peptides are frequently used for this purpose, both in eukaryotes and prokaryotes. In Gram-positive bacteria, these signals are either detected at the cell surface by histidine kinases or they are sensed inside the cell by RRNPP-type receptors after uptake by oligopeptide permeases[1,2]. When signals bind reversibly to receptors on the cell surface, the extracellular concentration of signaling molecules is the primary source of information, and the affinity of the receptor for its ligand controls signal transduction. In contrast, when signaling molecules are irreversibly pumped into the cell to activate an intracellular receptor, the extracellular signal concentration may not correlate with the concentrations sensed by the receptor. Thus, it is far from obvious what kinds of information cells can extract with the help of these networks to coordinate population-level behavior.

There has been tremendous progress in elucidating the molecular organization of the RRNPP signaling networks in recent years. RRNPP systems are widespread among *Firmicutes* and regulate traits which are commonly controlled by bacterial communication, such as cell differentiation, various forms of horizontal gene transfer, and the synthesis of (exo)factors that shape the interactions of these bacteria with other microbes and their hosts[2,3]. Binding of the signaling peptide to the receptor induces a conformational change that alters the activity of the receptor's output domain(s), which, depending on the receptor subtype, is either a DNA-binding domain or a protein-interaction domain or both[4–6]. Thus, some systems control gene expression directly, others indirectly, and a few do so in both ways. However, all systems share a common feature—namely, that the signals are produced by an export–import circuit. Cells express precursor peptides, which are subsequently secreted and cleaved by different proteases to produce the mature signaling peptides. These signals are then actively pumped into the cells by the conserved oligopeptide permease Opp[7,8], an ABC-type transporter that hydrolyzes ATP to drive the import of short oligopeptides[9]. Thus, RRNPP signaling networks represent prototypes for "pump–probe" networks, since signals are first "pumped" into the cell before they are "probed" (interpreted) by the respective RRNPP-type receptors.

The systems-level functions that are performed by these signaling networks are still unclear. They are commonly thought to facilitate "quorum sensing" in a bacterial population, i.e., the population-wide coordination of gene expression in response to changes in cell density[10,11]. Theoretically, they could indeed function as sensitive devices for cell-density monitoring[12], but whether RRNPP signaling networks actually implement a "quorum-sensing" type of regulation has been questioned[13,14]. They have also been hypothesized to function as timers for (multi-)cellular development[15–17], to coordinate the development of cellular subpopulations[18,19] and, under certain conditions, signaling could be self-directed and act in *cis* rather than in *trans*[20]. It is indeed conceivable that the more complex pump–probe network architecture allows for different types of extracellular information processing. However, this has not been systematically investigated.

One of the best characterized pump–probe network is the PhrA-RapA-Spo0F signaling pathway in *Bacillus subtilis*. The Rap proteins represent evolutionarily ancient RRNPP-type receptors[21] that are found in many *Bacilli*[22]. RapA and several other Rap homologs control the initiation of endospore formation by modulating the flux of phosphoryl groups through the sporulation phosphorelay to the master regulator Spo0A[23]. RapA binds to the response regulator Spo0F and stimulates the auto-dephosphorylation of Spo0F[23,24]. PhrA, the hydrophilic linear pentapeptide ARNQT that is derived from the *phrA* gene, binds to the RapA receptor at an allosteric site[14,15,25]. This induces a conformational change, which alters the interaction of the RapA with its response-regulator target Spo0F[4,24–26]. The *rapA-phrA* operon is highly regulated[27–32] and is

activated under both non-sporulating[30,31] and sporulating[16,27,28,33] conditions, indicating that signaling takes place in different situations. Interestingly, under some conditions the operon is expressed heterogeneously across the population[18] (a phenomenon that has been observed for other *rap-phr*-signaling systems[16]), which might point at a signaling function beyond classical quorum sensing[34]. Specifically, in a heterogeneous population its composition might be a relevant parameter for cellular decision-making.

Here we ask what regulatory functions pump–probe networks serve. To answer this question, we employ a combination of theoretical modeling of generic pump–probe networks and specific experiments on the PhrA-RapA-Spo0F pathway under non-sporulating conditions. We use Förster (fluorescence) resonance energy transfer (FRET) to monitor changes in the interaction of the RapA receptor with its response-regulator target Spo0F upon extracellular stimulation of *B. subtilis* cells with PhrA. We show that, in theory, pump–probe networks can exhibit different sensory modes that could mediate diverse functions, including quorum sensing, as well as chronometrically and ratiometrically controlled modes of regulation. The experimentally determined signal processing characteristics of the PhrA-RapA-Spo0F pathway suggest that the system could have evolved to sense the fraction of signal-producing cells in a heterogeneous population. We therefore propose that pump–probe networks could play an important regulatory function in coordinating decision-making in mixed populations.

## Results

**Pump–probe networks could serve different functions.** The characteristic pump–probe architecture that RRNPP-type networks employ for information processing distinguishes them from other bacterial communication systems. The defining features of a pump–probe network are that cells "pump" extracellular signaling molecules into the cytoplasm, effectively converting the extracellular into an intracellular signal, which is then "probed" (interpreted) by the appropriate intracellular receptor and transduced into a cellular output (Fig. 1a). We first asked whether pump–probe networks

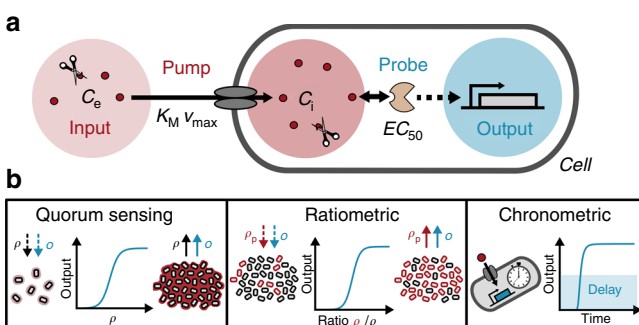

**Fig. 1 Schematics of a pump–probe network and its proposed regulatory functions. a** Schematics of the pump–probe model. Signals (red circles) are pumped into the cell, where they are probed (bound) by an intracellular receptor and transduced into an output. The conversion of extracellular to intracellular signal concentrations depends on signal transport by the pump, as described by Michaelis–Menten kinetics, and signal degradation as depicted by the scissors symbol. Intracellular signal transduction from the receptor to the output is modeled by a Hill function. See "Methods" for details. **b** Regulatory functions performed by pump–probe networks. From left to the right: *Quorum-sensing control*: The output $O$ is regulated in accordance with changes in cell density $\rho$. *Ratiometric control*: The output is regulated in accordance with changes in the composition of the population. Only a subset of cells (red, $\rho_p$) produces the signal, while all cells ($\rho_c$) in the population can take it up. The output is regulated by changes in the fraction of producing cells $f = \rho_p/\rho_c$. *Chronometric control*: Cells switch the output after a delay time that depends (mainly) on cellular parameters, independent of the social context.

could perform the regulatory functions that have been attributed to them (Fig. 1b).

With the help of a theoretical model described in detail in "Methods", we studied how information about the social context of a cell is encoded in the extracellular signal concentration $C_e$, and then converted into an intracellular signal $C_i$ that is transduced into a cellular output $O$. In brief, we consider an exponentially growing population of cells in which a fraction $f$ of cells produces the signal at rate $\pi$, while all cells take up the signal at a rate $v$. Indeed, our model suggests that pump–probe networks could enable the receptor to read out different kinds of information from the environment and perform different control functions, including quorum sensing, chronometric and ratiometric control (Fig. 2).

For example, in a population that produces more signaling molecules than can be removed by the cells (i.e., there is an effective net production rate $\pi_{eff} = f\pi - v_{max} > 0$), the extracellular concentration tracks changes in the population density ($C_e \sim \rho$). If the extracellular concentration is proportionally converted into an intracellular signal, the receptor can read out information about the population dynamics, and the output can be regulated in accordance with changes in cell density (quorum-sensing control, Fig. 2a). On the other hand, if the capacity for signal uptake exceeds signal production (i.e., $\pi_{eff} < 0$), the extracellular

concentration reaches a steady state that depends only on the fraction of signal-producing cells $f$ (fractional sensing), and not on cell densities. In this case, the output will be able to respond to changes in the population structure $f$ (ratiometric control, Fig. 2b). Finally, under conditions where signal accumulates so quickly as to saturate signal import, the intracellular concentration approaches a steady state at a rate that is (largely) independent of the social context of the cell and depends on cellular parameters only. As a result, a cell could delay an output for a specific time $\tau_{delay}$ (chronometric control, Fig. 2c). We thus conclude that pump–probe networks could perform various control functions, depending on network parameters and operating conditions. However, the network parameters of real systems are not well defined.

**PhrA alters FRET between CFP-RapA and YFP-Spo0F in *B. subtilis*.** To experimentally investigate signal processing by the PhrA-RapA-Spo0F pathway, we utilized a genetically encoded RapA-Spo0F FRET reporter (Fig. 3a). As shown below, this reporter provides direct readout of PhrA-induced changes in the RapA-Spo0F signaling complex at least under non-sporulating conditions, and thus of signaling activity within the cell. FRET, which relies on the distance- and orientation-dependent transfer

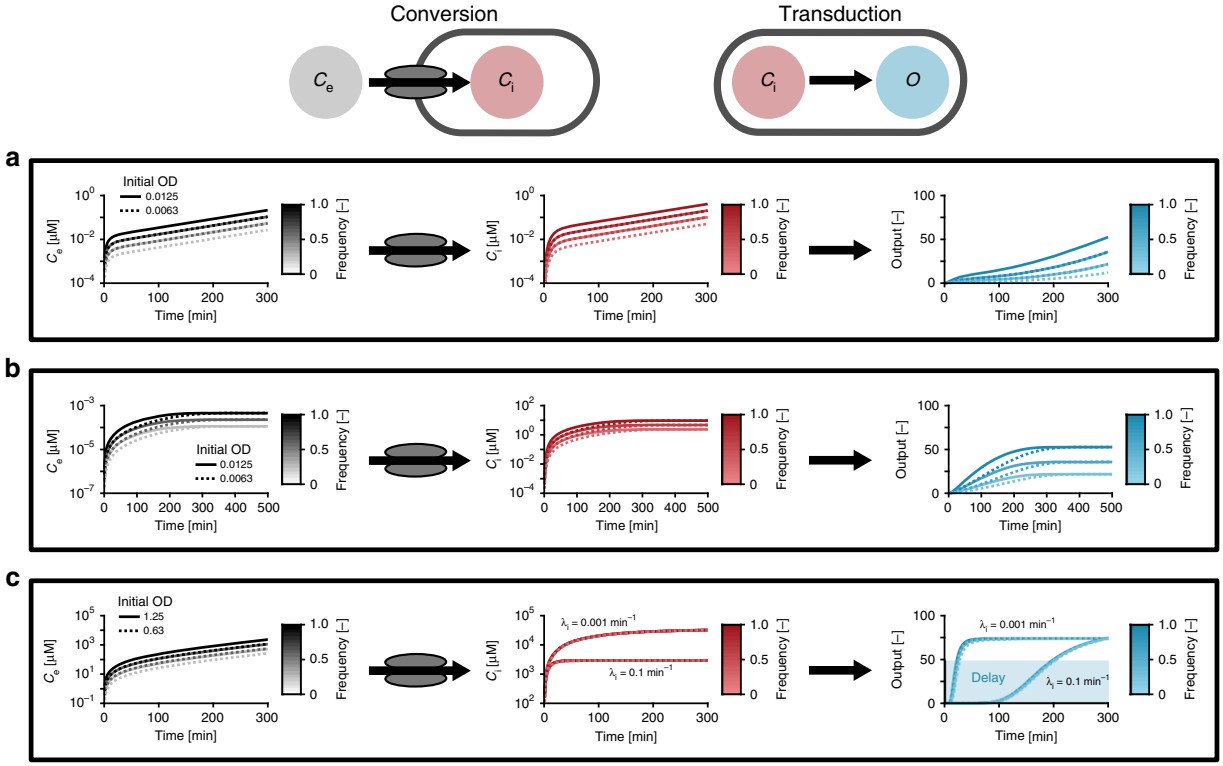

**Fig. 2 Pump–probe networks could execute various regulatory functions.** The results from simulations of the pump–probe model. A population of cells, suspended in a volume $V_e = 10$ mL, grows exponentially at a rate $\mu = 0.55$ h$^{-1}$. The initial population size is given by the inoculum $N_0 = \alpha \times$ OD$_{600\,nm}$, where $\alpha = V_e$ (mL) $\times 1.19 \times 10^8$ cells mL$^{-1}$. Starting at $t = 0$, a fraction of cells $f$ produces the signal at a rate $\pi$. The results are shown for an initial OD$_{600\,nm} = 0.0125$ (solid lines) and OD$_{600\,nm} = 0.0063$ (dotted lines) and for a homogenous ($f = 1$) and two heterogeneous populations ($f = 0.5, 0.25$, line color set by to the color bar). Active uptake ("pumping") of the signals into the cells converts the extracellular signal $C_e$ into an intracellular signal concentration $C_i$, which is probed by the receptor and transduced into an output $O$. **a** *Quorum sensing control*: $C_e$ and $C_i$ continue to rise as the population grows. The output tracks changes in the density of signal-producing cells, and varies both with the inoculum and $f$. Parameters: $K_M = 1.40$ mM, $v_{max} = 0.31$ amol min$^{-1}$, $\pi = 1$ amol min$^{-1}$, $\lambda_e = 0.1$ min$^{-1}$, $\lambda_i = 0.1$ min$^{-1}$; $n = 1$, EC$_{50} = 0.37$ μM. **b** *Ratiometric control*: Despite continuous population growth, the signal concentrations and the output approach a steady state that depends on $f$. Parameters: $K_M = 140$ nM, $v_{max} = 0.31$ amol min$^{-1}$, $\pi = 1$ zmol min$^{-1}$, $\lambda_e = 0$ min$^{-1}$, $\lambda_i = 0.1$ min$^{-1}$, $n = 1$, EC$_{50} = 8.5$ μM. **c** *Chronometric control*: $C_e$ rises and rapidly saturates uptake capacity. As a result, the accumulation of $C_i$ depends mainly on cellular parameters (i.e., transport $v_{max}$ and degradation $\lambda_i$). A switch-like receptor output enables the response to be delayed for a certain time, which is (largely) independent of the inoculum or $f$, but is tunable by cellular parameters, e.g., the peptide-degradation rate $\lambda_i$. Parameters: $K_M = 140$ nM, $v_{max} = 0.31$ amol min$^{-1}$, $\pi = 10$ amol min$^{-1}$, $\lambda_e = 0$ min$^{-1}$, $\lambda_i = 0.1$ min$^{-1}$, $0.0001$ min$^{-1}$; $n = 10$, EC$_{50} = 29$ mM.

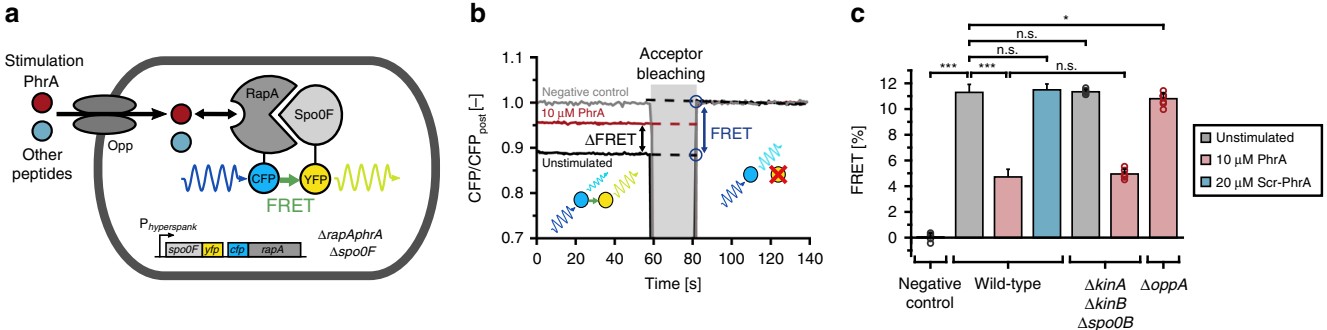

**Fig. 3 FRET assay used to study PhrA information processing in *Bacillus subtilis*. a** Scheme of a reporter cell that expresses CFP-RapA and Spo0F-YFP from an IPTG-inducible promoter, in a strain lacking the endogenous signaling components. When RapA and Spo0F form a signaling complex, intermolecular FRET between CFP and YFP occurs. Upon uptake of PhrA or other peptides by the oligopeptide permease Opp, PhrA binds to RapA and thus perturbs the interaction of RapA with Spo0F, thereby altering FRET. **b** Representative CFP trajectories from acceptor-photobleaching experiments with populations of unstimulated reporter cells (black line) and reporter cells stimulated with 10 μM PhrA for 6 min (red line). A FRET-negative control expressing free YFP and CFP is included for reference (gray line). YFP was bleached for 20 s to abolish FRET, which resulted in a corresponding increase in CFP emission. The FRET efficiency was calculated from the ratio of CFP emission before (CFP$_{pre}$) and after (CFP$_{post}$) photobleaching (blue circles), as determined by linear fits to the data (dashed lines). A relative increase in CFP emission that exceeds 0.6% is scored as significant, i.e., as indicating level of donor–acceptor FRET. **c** Barplot of FRET efficiencies derived from peptide stimulation experiments from left to the right: a FRET-negative control (BIB138), wild-type reporter cells (BIB625), a *kinA kinB spo0B* mutant (BIB1993) and a *oppA* mutant (BIB1563). Error bars: mean ± SD. Numbers of independent experiments: $n_e$ = 160, 112, and 21 for unstimulated, PhrA-treated and Scr-PhrA-treated (a scrambled version of the PhrA pentapeptide) in strain BIB625, respectively. $n_e$ = 5 (BIB138), $n_e$ = 6 (BIB1993), $n_e$ = 8 (BIB1563), respectively. Unpaired *t* test: n.s.: $P > 0.05$, ***$P < 0.001$. See Supplementary Data 1 for further statistical information. Source data are provided as a Source Data file.

of energy from an excited donor fluorophore to an acceptor fluorophore, has emerged as a powerful tool with which to study the function of bacterial signaling networks by monitoring protein–protein interactions in vivo[35]. When signaling alters the interaction between two fluorescently labeled proteins, changes in intermolecular FRET provide specific, fast, and quantitative readout of signaling activity. We measured FRET using acceptor photobleaching, where photoinactivation of the acceptor suppresses quenching of the fluorescence emitted by the donor, in proportion to the level of FRET observed prior to bleaching (Fig. 3b). This approach provides an absolute measure of the FRET efficiency, as the percentage change in donor fluorescence upon bleaching (see "Methods" Eq. (5)), which facilitates direct data comparisons across experiments.

We constructed a FRET reporter using CFP-RapA and Spo0F-YFP (Fig. 3a) in a *B. subtilis* strain that lacked the endogenous signaling genes (*ΔrapA-phrA Δspo0F*). The cells can be induced to express these stable (Supplementary Fig. 1a) and at least partially functional (Supplementary Fig. 1b –g) fusion proteins from an ectopic locus in the chromosome. Reporter cells were induced in S7$_{50}$ media and grown to a moderate cell density (optical density at 600 nm (OD$_{600 nm}$) ~1.6). Under these conditions, the PhrA-RapA signaling pathway is active in wild-type cells (as judged from activation of P$_{rapA-phrA}$[30,31] and the presence of PhrA in culture supernatants, as is shown in Supplementary Fig. 8), yet sporulation is inhibited. Acceptor-photobleaching experiments were performed on populations following a procedure that was previously established for *E. coli*[36,37], where integral CFP fluorescence of several hundred reporter cells is measured using a photomultiplier tube (Supplementary Fig. 2a). For the FRET reporter strain, an increase in CFP fluorescence indicative of FRET was observed upon bleaching of the acceptor (Fig. 3b, black line). In contrast, essentially no change in fluorescence was observed in a negative control expressing free cytoplasmic monomeric YFP and CFP (Fig. 3b, gray line; Supplementary Fig. 2b), or in a strain expression only CFP-RapA (Supplementary Fig. 2b), suggesting that any nonspecific contributions to FRET, e.g., from photoconversion or molecular crowding, are negligible. The FRET efficiency in unstimulated reporter cells was (11.2 ± 0.6)%, ±

indicating the standard deviation. This was comparable with (27.7 ± 2.1)% observed for a positive control (genetically fused YFP and CFP), while the spurious signal from the negative control was (0.04 ± 0.3)% (Fig. 3c, first and second bar; Supplementary Fig. 2b).

We then stimulated a population of reporter cells by adding 10 μM PhrA to the medium—a concentration that is sufficient to complement the sporulation defect of a signal-deficient *phrA* mutant[14,15] and to induce sporulation in the FRET reporter strain (Supplementary Fig. 1g). This resulted in a strong decrease in the FRET efficiency to (4.7 ± 0.6)% (Fig. 3b, red line and Fig. 3c, third bar). In contrast, addition of a sequence-scrambled pentapeptide (Scr-PhrA) did not alter FRET (Fig. 3c, fourth bar). Since the PhrA-RapA-Spo0F pathway is embedded in a complex signaling network, we analyzed whether FRET was affected by cross talk with other Phr signaling systems and whether changes in FRET arise as an indirect consequence of perturbations to phosphosignaling via the sporulation phosphorelay. Reporter cells neither responded to any other non-cognate Phr peptide (Supplementary Fig. 3a) nor the deletion of *kinA*, *kinB*, and the phosphotransferase *spo0B* affected the FRET efficiency of stimulated or unstimulated cells (Fig. 3c, fifth and sixth bar; Supplementary Fig. 3b). Together, these experiments show that the FRET reporter provides a specific and quantitative readout of signaling activity in the PhrA-RapA-Spo0F signaling pathway.

**Cells respond quickly, but activated cells recover slowly**. To characterize the dynamic properties of signal processing, we studied the response to the addition and removal of the extracellular PhrA. To this end, we applied a non-saturating PhrA stimulus and measured FRET by removing the cells from the medium at specific time points after stimulation $t_s$. We concomitantly monitored the depletion of PhrA from the medium by exposing fresh reporter cells to the spent supernatant for a defined period of time (Fig. 4a, see "Methods" for details). Within a few minutes of exposure to medium containing the stimulus (10 nM), the FRET efficiency decreased to (7.3 ± 0.7)%, while the extracellular PhrA levels declined to a level below the detection

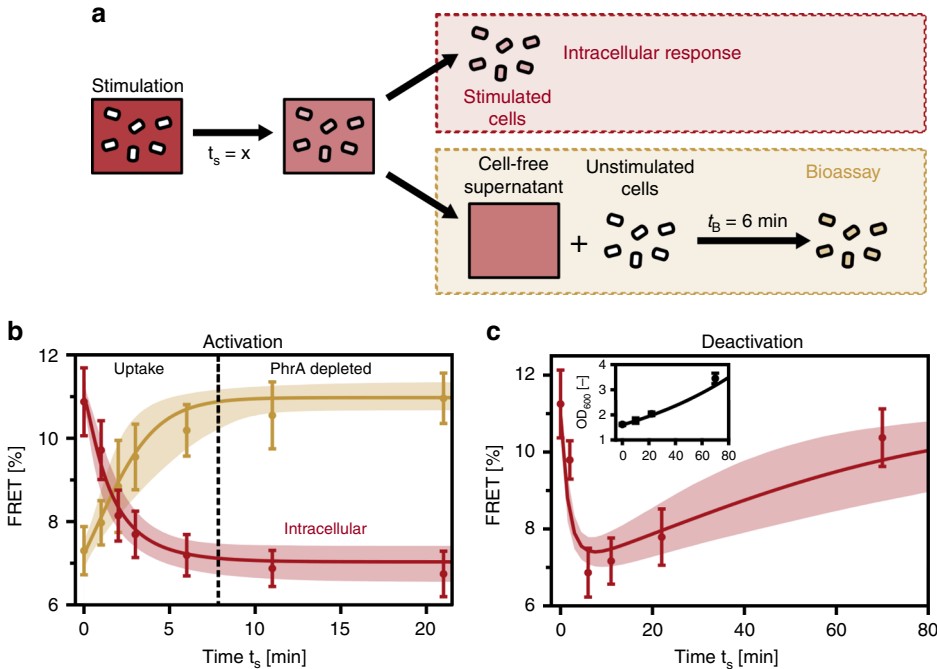

**Fig. 4 PhrA activates cells quickly, but cells recover slowly. a** Experimental setup used to characterize the signaling dynamics. Cells are exposed to a PhrA stimulus (red color) and take up the signal. Time $t_s$ refers to the time between addition of the stimulus and removal of cells from the medium. Stimulated cells and supernatants are separated to analyze the FRET response of stimulated cells and to measure the amount of PhrA remaining in the supernatants with the help of a bioassay (see "Methods"). **b** Activation dynamics upon stimulation with 10 nM PhrA. In stimulated cells (red), FRET decreases as a function of $t_s$ while, according to the bioassay, FRET values concomitantly rise to approach the levels seen in the unstimulated control, indicating that PhrA is depleted from the medium (yellow). Data: mean ± SD from $n_e = 9$. Red and yellow lines (shaded areas) depict the best fits (95% confidence intervals) to the pump–probe model (see Fig. 6). **c** Deactivation dynamics: FRET response to a non-saturating stimulus (10 nM) that was added to a growing population of cells at an optical density of $OD_{600\,nm} = 1.6$. Cells were removed from the culture at times $t_s$ to measure FRET. The red line (shaded area) depicts the best fit (95% confidence intervals) to an extended pump–probe model that considers effects of both population growth and intracellular signal degradation (see Fig. 6). Inset: Corresponding $OD_{600\,nm}$ curve. Data: mean ± SD, $n_e = 4$. Black line: fit to exponential population growth with rate $\mu = 0.58\,h^{-1}$. Source data for all panels are provided as a Source Data file.

limit of the bioassay (Fig. 4b). Notably, after extracellular PhrA had been depleted, the intracellular response was nevertheless sustained. Upon removal of external PhrA by resuspending the stimulated cells in signal-deprived medium and incubating them at 37 °C in microtubes, cells retained their activated state for 3 h (Supplementary Fig. 4a). However, cells also failed to grow under these conditions. We thus performed another stimulation experiment by adding PhrA directly to a shake-flask culture. As before, the growing cells rapidly responded to the addition PhrA (10 nM), resulting in a sharp drop in the FRET efficiency. This was followed by a slow increase in FRET over time (Fig. 4c; Supplementary Fig. 4b). Deletion of the *pepF* gene, an intracellular peptidase that is known to be capable of degrading PhrA when overexpressed[38], had little effect on the FRET response (Supplementary Fig. 5a), suggesting that other peptidases may contribute to signal degradation. Also, in cell-free supernatants, the extracellular PhrA (10 nM) was found to remain stable for hours (Supplementary Fig. 5b).

**Competition for substrate uptake inhibits PhrA signaling.** When we deleted the gene for the oligopeptide-binding protein OppA that delivers the peptides to the Opp transporter[7,39], there was virtually no response, as expected (Fig. 3c, seventh bar).

We then measured the response starting from different initial extracellular concentrations $C_e$ in the absence and presence of a competing peptide. With increasing signal concentrations, FRET gradually decreased and then levelled out at $(4.7 ± 0.6)\%$ (Fig. 5a). Scr-PhrA strongly inhibited the PhrA-mediated response when

the competing peptide was present in excess (Fig. 5b). However, adding scr-PhrA to cells prior to stimulation with PhrA had no effect, indicating that the peptide competes with PhrA for uptake by Opp, but not for RapA receptor binding (Fig. 5c).

**Signal processing is well described by the pump–probe model.** The signal processing characteristics of the PhrA-RapA-Spo0F pathway are jointly determined by its signal conversion and signal transduction properties. With the help of the pump–probe model, one should be able to disentangle the two and learn how extracellular signals are converted into an intracellular signal and how cytosolic PhrA then affects the RapA-Spo0F signaling complex (Fig. 6a). In order to quantitatively describe our data, we investigated signal processing by the pump–probe model assuming that the FRET response is governed by the intracellular PhrA concentration and described by a Hill function. We included substrate competition, assuming that the Opp pump transports all penta-peptides with the same efficiency (see "Methods" for details). Furthermore, on the short timescale of our activation experiments in Figs. 4b and Fig. 5, cell growth and signal degradation are negligible (Supplementary Figs. 4 and 5). For the long-term response dynamics of growing cells (Fig. 4c), both intracellular signal loss from dilution due to cell growth at the experimentally determined rate (inset Fig. 4c) and linear signal degradation were explicitly modeled. We then fitted our data set to the pump–probe model, which resulted in excellent agreement (all lines in Figs. 4 and 5) given the parameters summarized in Table 1.

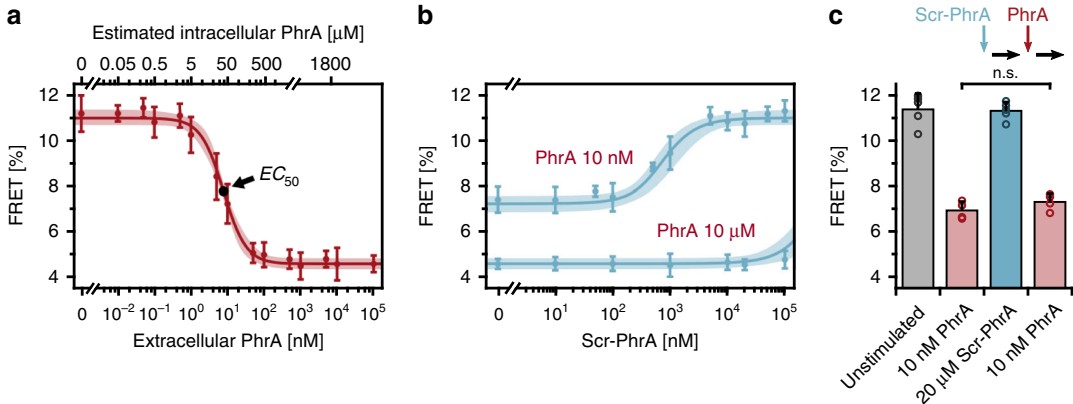

**Fig. 5 Competition for substrate uptake inhibits PhrA signaling. a** Response to increasing PhrA levels ($t_s = 6$ min). Data: mean ± SD from $n_e = 10$. The curves (shaded areas) depict the best fits (95% confidence intervals) to the pump–probe model (see Fig. 6). The top axis denotes the intracellular PhrA concentrations as estimated by the model. **b** Competition for peptide uptake inhibits PhrA signaling. FRET response curve of reporter cells stimulated with 10 μM and 10 nM PhrA in the presence of the indicated concentrations of a competing peptide, scrambled PhrA ($t_s = 6$ min). Data: mean ± SD from $n_e = 5$. Lines and shaded areas denote best fit and the 95% confidence interval to the pump–probe model. **c** Barplot of FRET values derived from a sequential stimulation experiment, including individual data points. Prior exposure of cells to 20 μM Scr.-PhrA did not alter the response to 10 nM PhrA. Data: mean ± SD from $n_e = 4$. Unpaired $t$ test: $P$ (0.21) > 0.05 (n.s.). See Supplementary Data 1 for further statistical information. Source data for all panels are provided as a Source Data file.

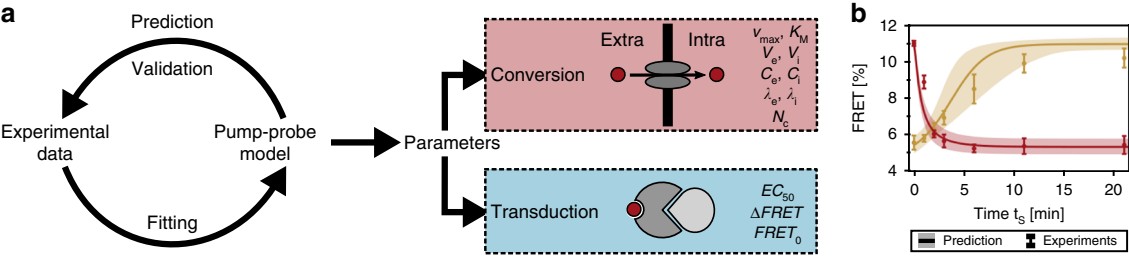

**Fig. 6 PhrA signal processing is well described by the pump–probe model. a** Experimental–theoretical workflow. Experimental data were fitted to the pump–probe model to infer the parameters that determine extra- to intracellular signal conversion and transduction of the signal into a FRET output. The inferred parameters were then used to make testable predictions and validate the model experimentally. See lines in Figs. 4 and 5 for best fits using parameters in Table 1. **b** Model validation: model-based predictions with 95% confidence intervals (lines and shaded area) and experimental results for a stimulation experiment with 30 nM PhrA. The response dynamics to stimulation (red) is shown, together with the depletion of PhrA from the supernatants as measured by the bioassay (black). Data: mean ± SD from $n_e = 3$. Source data are provided as a Source Data file.

## Table 1 Parameters of the PhrA signaling network in *Bacillus subtilis*.

| Parameter | Lower bound | Upper bound | Best fit |
|---|---|---|---|
| $v_{max}$ (mol × $10^5$ min$^{-1}$) | 1.2 | 18 | 1.9 |
| $K_M$ (nM) | 20 | 260 | 140 |
| $EC_{50}$ (μM) | 32 | 46 | 38 |
| $FRET_0$ | 0.107 | 0.113 | 0.110 |
| $\Delta FRET$ | 0.060 | 0.069 | 0.064 |
| $n$ | 1.1 | 1.8 | 1.4 |
| $\lambda_i$ (h$^{-1}$) | 0.14 | 1.58 | 0.63 |

The parameters of the pump–probe model were estimated fitting all experimental results from Figs. 4 and 5. The lower and upper bound correspond to the 95% confidence interval (see "Methods").

To further demonstrate that the simple pump–probe model adequately describes PhrA signal processing, we predicted and experimentally verified the extra- and intracellular response dynamics to a 30 nM PhrA stimulus, which resulted in very good agreement (shaded areas depict 95% confidence intervals of the model prediction in Fig. 6b). In addition, the response to a higher 100 nM stimulus was also captured satisfactorily (Supplementary Fig. 6). We thus conclude that PhrA signal processing is well described by the pump–probe model.

**Signal conversion results in strong signal amplification.** Based on the inferred parameters and their 95% confidence intervals listed in Table 1, we can provide more details in the signal-transduction process. First, our model suggests that FRET between RapA and Spo0F changes in a graded manner in response to increasing concentrations of intracellular PhrA and then saturates at a finite level. Thus, signal transduction is well approximated by a simple hyperbolic response function (best fit: $n = 1.4$), indicating that there is little, if any, cooperativity present in signal transduction. Second, the inferred $EC_{50}$ (best fit: 38 μM) suggests that relatively high intracellular signal concentrations are required for signal transduction. To suppress FRET to half-maximum, a ~1000-fold lower extracellular signal concentration (nM) than the intracellular $EC_{50}$ ~μM was required (Fig. 5b). This strong signal amplification upon extra- to intracellular signal conversion is the consequence of active and efficient signal transport by the Opp pump, which allows the accumulation of PhrA in the small cell volume against an external concentration gradient. Finally, the inferred characteristic timescale for signal processing $\tau = 1/(\mu + \lambda_i) \sim 50$ min is relatively long, and signal degradation ($\lambda_i$) and the dilution rate due to cell growth ($\mu$) each contribute roughly equally. As a consequence, the intracellular concentration tracks fluctuations in extracellular concentrations on timescales faster than $\tau$ rather poorly. Instead, cells integrate

extracellular signals over the characteristic signal processing time τ—or shorter times—until all signals are depleted from the medium (Fig. 4c).

**Population of cells process PhrA in a dose-dependent manner.** When cells compete with each other for signal uptake, the signal conversion depends on the cell density in addition to the initial

extracellular concentration. Both factors can be combined into a single environmental parameter, the signal dose, defined as the amount of available signaling molecules per cell. We thus investigated to what extent the response to PhrA stimulation depends on either factor alone (Fig. 7a, b) and the combined effect as described by the dose, respectively. To this end, we kept the dose at a fixed level and varied the extracellular signal concentration and cell density, respectively. The response curves

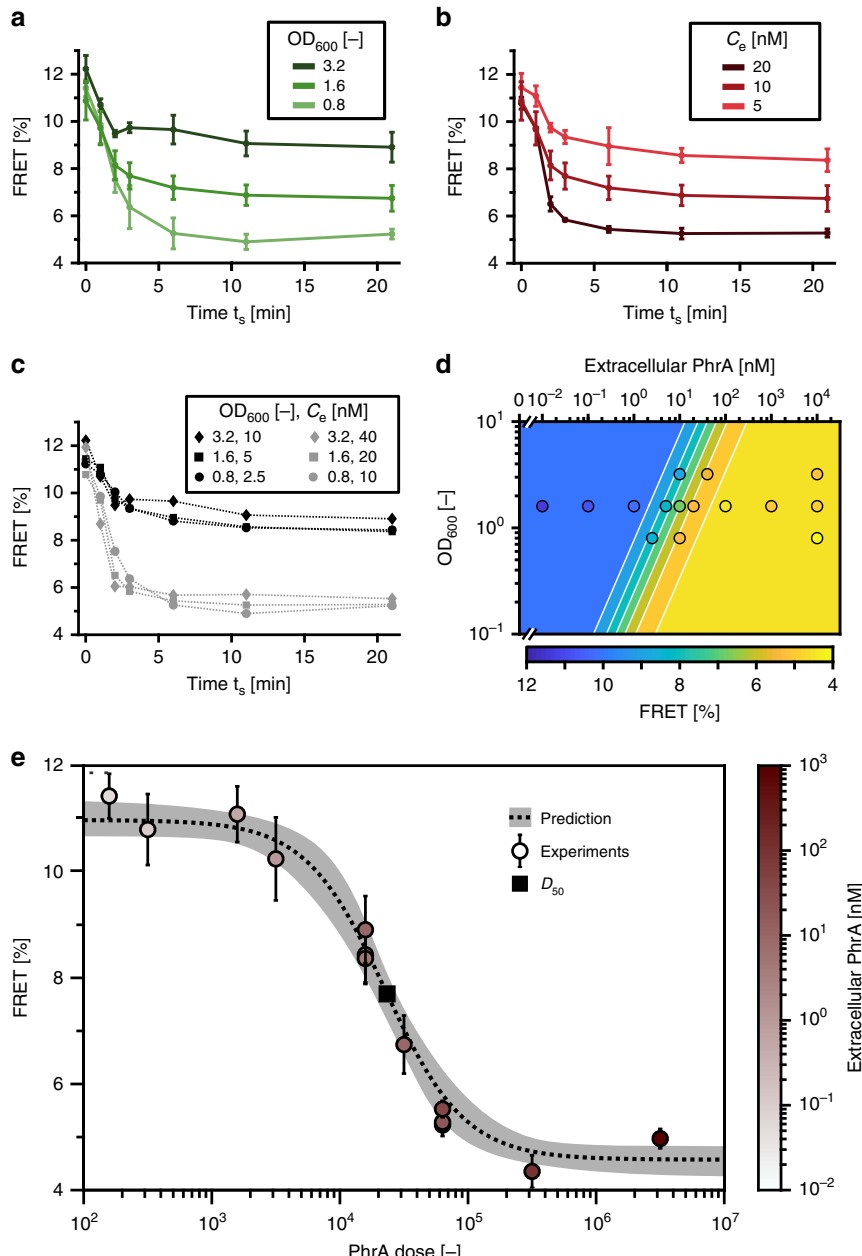

**Fig. 7 The response to PhrA depends on the signal dose. a** The FRET response of the population varies with cell density. Time course of stimulation with 10 nM PhrA for three different cell densities. Data: mean ± SD from $n_e = 3$. **b** The FRET response varies with the extracellular concentration when cell density is fixed (OD$_{600}$ ∼1.6). Data: mean ± SD from $n_e = 3$. **c** Response curves using the indicated extracellular concentrations and cell densities, respectively. Proportional changes in both number of cells and extracellular concentrations, i.e., exposure of cells to the same dose, lead to very similar response curves. The response curves from different doses are distinct. **d** The contour plot of the lowest level of FRET as a function of cell density and the extracellular concentration of PhrA as predicted by the pump–probe model shows that variations in FRET require a change in the signal dose $D$. Filled circles denote experimental data. Color of circles indicates measured FRET values as set by the scale of the color bar on the bottom. **e** Dose–response curve: all of the minimal FRET values obtained from stimulation experiments with different extracellular concentrations collapse onto the predicted dose–response curve from the pump–probe model with a $D_{50} = 2.4 \times 10^4$ molecules. Color of circles depict extracellular concentrations used in the experiment according to the scale of the color bar on the right. Source data for all panels are provided as a Source Data file.

obtained with different doses were clearly distinct, especially with respect to the final degree of FRET inhibition achieved (Fig. 7c). According to the model, response curves corresponding to the same dose will all converge on the same final output, although the kinetics varies. Thus, we next predicted the maximal degree of inhibition as a function of the extracellular concentration and the cell density, respectively, from the pump–probe model, and again found excellent agreement with our experimental data (Fig. 7d). Indeed, all our data collapsed onto a single dose–response curve that describes how FRET is inhibited as a function of the number of extracellular signaling molecules per cell and fits the prediction from the model very well (Fig. 7e). The dose for a half-maximal response is $D_{50} = 2.4 \times 10^4$ PhrA molecules. This indicates that "dose" is in fact the dominant environmental factor that determines signal processing under our conditions.

**The PhrA-RapA-Spo0F system is capable of ratio sensing**. The dose-dependent response provides evidence for the capacity for ratiometric output control in a heterogeneous population, where all cells take up peptides but only a fraction $f$ of the population produces the signal. During the signal integration time $\tau$ each "producer" cell synthesizes $N_{out} = \pi \tau$ PhrA molecules, where $\pi$ is the PhrA production rate. Since only a fraction $f$ of the population produces the signal, the number of available PhrA molecules per cell, i.e., the dose, is given by $D - fN$. Therefore, changes in the fraction $f$ of producers result in a change in the signaling output, provided the intracellular pathway is not saturated. At steady state, signal production and signal uptake balance each other; thus one can estimate $D$ from the number of signaling molecules taken up by each cell during the signal integration time, i.e., $D \sim N_{in} = \frac{\tau v_{max} C_e}{C_e + K_M}$. We thus next estimated the PhrA concentration in the supernatant of wild-type cells with the help of a sensitized bioassay, and found that it is present at sub-nM concentrations, $C_e \sim 0.4$ nM (Supplementary Fig. 7). Hence, using the inferred parameters from Table 1, we estimate $D = 2.7 \times 10^4$, which is comparable with the dose required to induce a half-maximal response $D_{50} = 2.4 \times 10^4$ in our stimulation experiments. We thus conclude that the parameters of the PhrA signaling system are properly balanced to facilitate ratiometric output control in heterogeneous populations.

## Discussion

Cellular signaling systems based on RRNPP receptors have emerged as promising targets for manipulating the behavior of bacterial populations in diverse biotechnological and biomedical settings[2]. Our systems-level analysis of signal processing provides key insights into both the functioning of pump–probe networks and the signal conversion and transduction properties of the prototypical *rapA-phrA* system in *B. subtilis*. By utilizing a novel FRET reporter, we could quantitatively study important features of signal processing, which has enabled us to infer network parameters with the help of the pump–probe model. The model fits the experimental data very well (Figs. 4 and 5), and it has predictive power (Fig. 6b). For high signal concentrations (100 nM), additional effects could come into play (Supplementary Fig. 6), but these should have little relevance because PhrA levels in supernatants were orders of magnitude lower (Supplementary Fig. 7). However, we add the caveat that our model assumes that FRET changes as a function of the intracellular PhrA concentration, which implies that receptor kinetics is fast relative to all other processes. This is a reasonable assumption, given that the activation dynamics is limited by signal uptake and the $K_d \sim \mu M$[40] for Rap-Phr interactions is relatively high. Finally, the parameter values inferred from our model are generally in good agreement with previous data on Opp-based transport in

*B. subtilis*[41] and PhrA signal transduction[25], further increases our confidence in our model.

The combined experimental and theoretical approach allows us to provide further insights into the individual processes that govern pump–probe signaling. The Opp pump imports $v_{max} = 1.9 \times 10^5$ molecules min$^{-1}$ at maximal speed; thus, cells clear peptides from their environment very efficiently. However, the cellular signal import rate under physiological conditions is much lower ~500 PhrA molecules min$^{-1}$ owing to the low PhrA concentrations in the medium. Thus, the peptide-binding protein OppA must have sufficient affinity to facilitate signaling at low peptide concentrations. Indeed, the inferred effective affinity of peptide transport ($K_M = 140$ nM) is about two orders of magnitude lower than that for OppA from *Lactococcus lactis*, which feeds on peptides in protein-rich environments[42,43]. In the presence of other peptides, competition for peptide uptake slowed down PhrA signal accumulation in *B. subtilis*, and thereby interfered with signaling. Notably, peptide-rich media have an inhibitory effect on RRNPP signaling[17,21,44] and in *Enterococcus faecalis* signal import does not occur via OppA but with the help of a signal-specific peptide-binding protein PgrZ[45], presumably to avoid such competition and to minimize signal interference.

After signals are pumped into the cell, the intracellular signal concentration is probed by RRNPP-type receptors. Raps belongs to a subclass of RRNPP receptors termed switchable allosteric modulator proteins (SAMPs)[46], because they modulate the activity of response regulators; Phr peptides switch this interaction by binding to the receptor in a 1:1 stoichiometry at an allosteric site[4,40], and structural studies suggest competitive allosteric inhibition as the dominant mode of signal transduction[4,40]. While the analysis of receptor function in vitro is very advanced, functional in vivo analyses have lagged behind. Thus far, receptor function has been assessed rather indirectly, using gene expression[4] or cell differentiation readouts[38]. Moreover, the responses are typically reported as a function of (initial) extracellular concentration, which may not correlate well with the intracellular signal concentration that is detected by the receptor. With the help of the pump–probe model, and utilizing FRET to directly probe the interaction of the receptor with its response-regulator target, which allowed us to infer the intracellular signal concentrations (Fig. 5b), we have successfully met these challenges. Indeed our data suggest that extra- and intracellular concentrations are very different. PhrA signaling operates at very low extracellular signal concentrations (sub-nM), although the intracellular signal transduction exhibits limited sensitivity, as indicated by the fairly high $EC_{50}$ ($\mu M$). Given the relatively low nM–$\mu M$ extracellular signal concentrations in culture supernatants that have been measured in other systems[41,47] and the high $K_I$ values determined for other RRNPP receptors[25,40], such strong signal amplification upon extra- to intracellular signal conversion is probably quite common.

Our data also provide first insights into how the RapA receptor functions in the bacterial cell under non-sporulating conditions where PhrA signaling is active in wild-type populations. In line with expectations based on the 1:1 stoichiometry of PhrA binding to the RapA receptor, the FRET response mediated by the PhrA-RapA-Spo0F pathway is well described by a hyperbolic response function. If PhrA acted to dissociate the RapA-Spo0F complex, it should reduce, and eventually abolish, FRET. Indeed, PhrA inhibited FRET, but substantial FRET signal above the negative control still remained at high signal concentrations. This residual FRET is not an artefact of population measurements using acceptor photobleaching, as E-FRET measurements[48] on single cells also confirmed that all cells respond to PhrA but retain residual FRET

(Supplementary Fig. 8). Hence, the receptor–regulator complexes may not (fully) dissociate and instead, a stable ternary complex might form. In support of this inference, the in vitro action of PhrA on RapA is best described by a partial noncompetitive inhibition mechanism[25], which implies that PhrA-RapA-Spo0F complexes contribute to signaling. Thus upon activation, Raps may remain (partially) bound to their (unphosphorylated) response-regulator targets[24], which could fine-tune the cellular response to receptor stimulation[46]. Since our experiments were conducted under non-sporulating conditions, FRET likely reports on the interaction of RapA with unphosphorylated Spo0F. In vitro data suggest that phosphorylation of Spo0F alters and stabilizes the interaction with RapA[24]. This could affect FRET under sporulating conditions and should be investigated in the future.

In bacteria, there are numerous examples of different network architectures that utilize diffusible signaling molecules to regulate cellular behaviors[12,49]. Precisely what kind of information cells can extract with the help of these sensory networks remains under debate[50,51]. The most popular interpretation is that they are utilized for cell-density sensing[52]. In the case of RRNPP-based signaling, the receptors are commonly referred to as "quorum-sensing" receptors[4,6,53]. However, the experimental evidence that these systems mediate a cell-density-dependent type of regulation is—at least not only for the Rap systems in *B. subtilis*[14,54], but also others[17]—rather weak. However, the capacity for quorum sensing is in principle only one of several control functions that pump–probe networks could perform, as suggested by our model. It is thus possible that this or other types of regulation occur in other systems or under different conditions. For example, upon the transition from non-sporulating to sporulating conditions, the inferred parameter values for the PhrA-RapA network might change, since all signaling components (and their interactions) are regulated by a complex network[55]. In principle, this could switch the network's control function, e.g., from ratio to chronometric or quorum-sensing control, respectively.

Notably, at least under some physiological conditions, Phr signaling may function to coordinate cellular decision-making in the context of a heterogeneous population. Population heterogeneity could be phenotypic, as in the case of PhrA signaling under sporulation conditions, where only a subpopulation of cells that delays sporulation initiation and continues to divide upregulates the expression of the signaling system[18,19], or genetic, as in the case of Phr_LS20 signaling, where Phr_LS20 is expressed from a plasmid to regulate conjugation to other cells[56]. How signaling contributes to decision-making in heterogeneous microbial populations is very much understudied. In yeast, the pheromone pathway mediates sensing of the sex ratio to control cellular investments in mating[57]. Our experiments show that *B. subtilis* processes PhrA signals in a dose-dependent manner: the signaling output is determined by the level of extracellular signal per cell. This is a strong indication that pump–probe networks are capable of mediating fractional (ratiometric) population sensing in mixed populations without any additional regulation. Fractional population sensing requires cells that do not produce the signal to take up the signal. This is likely to be the case for signals that rely on nonspecific transport by the conserved oligopeptide permease Opp.

For pump–probe signaling networks, the capacity for fractional sensing is built into the basic network architecture. This contrasts with the case in yeast, where ratio sensing is performed by a membrane-bound receptor signaling pathway that requires specific additional regulatory features to perform this function[57]. We therefore propose that fractional sensing could be a widespread function of oligopeptide-based signaling involving the use of Opp pumps to coordinate the behavior of bacteria in mixed populations. It could also be exploited by selfish genetic elements (including plasmids[56] and integrative and conjugative elements (ICE)[58] and viruses[47] that are known to carry peptide-based pump–probe signaling circuits in their respective genomes.

## Methods

**Mathematical model for pump–probe networks.** We consider a population of cells that are homogenously distributed in a volume $V_e$. The population grows exponentially at a rate $\mu$, $N_c = N_c^0 e^{\mu t}$. A fraction of the population $f$ produces the signal at a constant rate $\pi$. If $f = 1$, the population is homogenous, and heterogeneous otherwise. At time $t = 0$, the extracellular signal concentration is $C_e^s$ ($t = 0) = C_{stim}$, and there is no signal inside the cell, i.e., $C_i^s$ ($t = 0) = 0$. Other peptides are present at concentration $C_e^o$ and compete for peptide import. Each cell imports peptides at a rate $r$, which is a function of the total peptide concentration $C_e = C_e^s + C_e^o$. Peptide uptake is assumed to occur with Michaelis–Menten kinetics at a maximum particle flux per cell of $v_{max}$, and approaches saturation with increasing $K_M$. As the signal is imported at rate $v$, the intracellular signal concentration $C_i^s$ rises at a rate $v/V_i$. Peptides are degraded extra- or intracellularly at rates $\lambda_e$ and $\lambda_i^s$, respectively and intracellular signals are also diluted by cell growth. The following set of ordinary differential equations describes how, for the intracellular signal concentration $C_i^s$, the extracellular signal concentration $C_e^s$, and the total extracellular peptide concentration $C_e$, change as a function of time:

$$\frac{dC_i^s}{dt} = v_{max} \frac{C_e^s}{K_M + C_e} \frac{1}{V_i} - (\lambda_i^s + \mu)C_i^s, \tag{1}$$

$$\frac{dC_e^s}{dt} = \pi f \frac{N_c}{V_e} - N_c v_{max} \frac{C_e^s}{K_M + C_e} \frac{1}{V_e} - \lambda_e C_e^s, \tag{2}$$

$$\frac{dC_e}{dt} = -N_c v_{max} \frac{C_e}{K_M + C_e} \frac{1}{V_e} - \lambda_e C_e. \tag{3}$$

We assume that receptor activation and signal transduction occur rapidly. In this case, the output $O$ becomes a function of the intracellular signal concentration and is modeled by a Hill function for simplicity:

$$O(t) = O_{max} \frac{C_i(t)^n}{EC_{50} + C_i(t)^n}. \tag{4}$$

Here $O_{max}$ is the maximal output, $EC_{50}$ is the intracellular peptide concentration that yields a half-maximal response, and $n$ is the Hill coefficient.

The model Eqs. (1)–(4) were solved numerically with Matlab R2017b (MathWorks Inc.) using the ode15s solver.

**Media.** Strains were grown in LB-media (Lennox version)[59] or S7_50 minimal medium[60,61] at 37 °C with aeration. Difco sporulation medium (DSM), growth medium (GM), and resuspension medium (RM) were prepared according to standard protocols[62]. LB agar plates were used to select transformants. When required, the appropriate antibiotics and amino acids were added as follows: for *E. coli*, ampicillin (100 μg ml$^{-1}$); for *B. subtilis*, spectinomycin (100 μg ml$^{-1}$), erythromycin (2 μg ml$^{-1}$), tetracycline (10 μg ml$^{-1}$), and tryptophan (50 μg ml$^{-1}$).

**Plasmid construction.** All plasmids and primers are listed in Supplementary Tables 1 and 2, respectively. *Escherichia coli* DH5α (Invitrogen, Carlsbad, CA, USA) was used for cloning. All plasmids were verified by sequencing.

*FRET reporter plasmids:* FRET reporters were constructed with the help of pDR111 by restriction-enzyme ligation cloning (RELC). *rapA* and *spo0F* were amplified from *B. subtilis* 168 genomic DNA and fused via a GSGGV linker to monomeric *yfp-venus* and *ecfp(Bs)*, respectively. We first constructed expression plasmids for the individual fusion proteins in order to test for their functionality in *B. subtilis*. *ecfp(Bs)* was amplified from pDR200, fused to the N-terminus of *rapA* by a joining PCR, and cloned into pDR111 by RELC using the enzymes NheI and SphI, resulting in EIB77. *yfp-venus* was amplified from AEC253, fused to the C-terminus of *spo0F*, and cloned into pDR111 by RELC using the SalI and NheI enzymes, resulting in EIB283. To obtain the FRET reporter plasmid, *spo0F-yfp* was excised from EIB283 with SalI and NheI, and ligated into EIB77 to generate EIB284. The FRET reporter contains an operon comprising the *spo0F* and *rapA* fusion protein genes under the transcriptional control of the IPTG-inducible P_hyperspank promoter.

For the FRET-negative control plasmid (EIB152), used to express free cytoplasmic YFP and CFP under the control of IPTG, both genes were cloned into pDR111 by RELC using the enzyme pairs SalI and NheI and NheI and SphI, respectively. For the FRET-positive control plasmid (EIB151), we fused *yfp-venus* to *cfp(Bs)* with a GSGGV linker by a joining PCR and cloned the product into pDR111 by RELC using SphI and NheI.

*Plasmids for gene deletions:* The plasmids for clean deletions were constructed by amplifying 500-bp fragments located upstream and downstream of the region of interest from genomic DNA. The DNA fragments were fused by PCR and cloned into the pMAD vector by RELC using the enzymes SalI and BglII.

*Plasmids for xylose induction of PepF:* The *pepF* coding sequence, including its native transcription terminator, was amplified from genomic DNA. As a ribosome-

binding site, the consensus Shine-Dalgarno sequence was added upstream of the start codon. The insert was cloned into pAX01[63] by RELC with the enzymes SpeI and BamHI.

**Strain construction.** All *B. subtilis* strains were derived from 1A700 (W168) and are listed in Supplementary Table 3.

*FRET reporter strain:* To construct the FRET reporter strain the *rapA-phrA* operon was deleted from W168 using plasmid EIB185, following a protocol similar to that of Arnaud et al.[64]. The resulting strain was subsequently employed to delete *spo0F* after transformation with plasmid EIB281 to yield strain BIB415 (Δ*rapAphrA* Δ*spo0F*) using the same protocol. At each step, we verified that the gene had been deleted from its chromosomal locus, that the pMAD plasmid had been lost and, finally, that the gene was entirely absent from the chromosome by appropriate PCRs (notably, we found that some transformants had acquired a gene copy in another locus; these were discarded). FRET reporter strains (BIB625 and BIB914) were obtained by transforming BIB415 with the indicated FRET reporter plasmid EIB282 according to standard protocols[62]. Correct integration of the reporter constructs at the *amyE* locus was verified by an *amyE*-negative phenotype, while appropriate PCRs were performed to verify the correct size of the integrated construct in the *amyE* locus and confirm that no additional single crossover had occurred (absence of the *ampR*-resistance cassette). We note that integrations carried out with pDR111 (and probably many other common *amy* integration vectors) can result in a ~250-bp deletion in the adjacent *ldh* locus. We therefore verified by PCR that our transformants retain an intact *ldh* locus[65]. In addition, we confirmed by PCRs the deletion of *rapAphrA* and of *spo0F* from the final strain. The FRET control strains BIB134 and BIB138 were obtained by transforming the wild-type strain with plasmids EIB151 and EIB152, respectively, and verified by PCRs.

*Mutant reporter strains:* Deletions of indicated genes (*kinA, kinB, spo0B, oppA,* and *pepF*) were made in the FRET reporter strain (BIB625) using pMAD-derived plasmids (Supplementary Table 2), and verified as described above. A xylose-inducible *pepF* construct was introduced into the *lacA* locus (BIB1612) by transforming BIB625 with EIB544. Correct integration in the *lacA* locus was verified by a PCR of the *lacA* locus and a PCR for the *ampR*-resistance cassette to confirm that no additional single crossover had occurred.

**Functionality of fluorescent fusion proteins.** Protein stability was assessed by western blotting[66]. Proteins were harvested from *B. subtilis* cells grown in 20 mL of LB medium and induced with IPTG. Fusion proteins were detected with anti-GFP conjugated with HRP antibody (Invitrogen, Catalogue no. A10260, lot number 898225). Function of fusion proteins was assessed by plating on DSM agar plates in combination with measurements of colony opacity using ImageJ[67]. Sporulation of FRET reporter cells was induced by the resuspension method[62] by applying a shift from GM to RM media with 10 μM IPTG and the indicated concentrations of PhrA. After 24 h of incubation at 37 °C in shake-flask culture, the sporulation frequency was determined by microscopy.

**Quantitative FRET assays.** *Induction of FRET reporters:* Reporter cells were inoculated from a single colony into 5 ml of LB medium with antibiotics and incubated on a rotary shaker (Infors HT Multitron) at 180 rpm and 37 °C for 7 h. Cells were resuspended at $OD_{600\,nm} = 0.003$ in 5 ml of $S7_{50}$ medium and incubated for 16 h overnight. Expression of the fusion proteins was induced by resuspending the reporter cells at an $OD_{600\,nm} = 0.04$ in 10 ml of fresh $S7_{50}$ supplemented with 100 μM IPTG in a 100-ml flask. When applicable, protein expression from a xylose-inducible promoter was induced by adding xylose at a final concentration of 1% (w/v). Cells were grown to a final $OD_{600\,nm}$ ~1.6.

*Stimulation of reporter cells with synthetic peptides:* To stimulate the reporter cells, we added 5 μl of an appropriate concentrate (purity >95%) of synthetically synthesized peptides (Peptides and Elephant, Henningsdorf, Germany) to 500 μl of the induced culture in a conical 2-ml reaction tube. After incubation for time $t_{inc}$, cells were centrifuged for 1 min at 17,000×*g*. Thus, after addition of the stimulus, cells spent $t_s = t_{inc} + 1$ min in the medium. If not otherwise specified, the incubation was performed at room temperature for 5 min without shaking, and thus $t_s = 6$ min. The pellet was washed by resuspending cells in 500 μl of phosphate-buffered saline (PBS). In case of subsequent stimulations, the washed reporter cells were resuspended in the culture medium, and a second round of stimulation was started as described above. Finally, the washed pellet was resuspended in 5 μl PBS and spread on an agarose pad (1% ultrapure agarose (Invitrogen) in PBS).

*Bioassay and supernatant analysis:* The supernatants were collected after centrifugation of stimulated cells. Aliquots ($V_S = 450$ μl) of the cell-free supernatants were mixed with 50 μl of a 10× suspension of fresh unstimulated reporter cells, for $t_B = 5$ min + 1 min (time for incubation plus centrifugation) at room temperature and then processed as described above. To detect extracellular PhrA in $S7_{50}$ cultures, wild-type cells (BIB224) were grown for 5 h to an $OD_{600\,nn}$ ~ 1.6 and pelleted by centrifugation. The supernatants were then filtered through a 0.2-μm PES membrane, and analyzed as described above with the following modification. To increase the

sensitivity of the bioassay, the volume of analyzed supernatant and the incubation time were increased to $V_S = 1950$ μl and $t_B = 20$ min + 1 min, respectively.

*Deactivation dynamics of stimulated cells (FRET recovery response):* The deactivation of the pathway after stimulation was studied using two assays. First, pre-stimulated cells were washed and resuspended in 500 μl of culture medium and incubated at 37 °C on a thermoshaker. Second, to monitor the recovery of FRET under growth conditions, PhrA was added directly to a shake-flask culture (9 ml) to a final concentration of 10 nM, and the response dynamics was followed over three hours by withdrawing 500 μl samples at the indicated times. In each case, an unstimulated population served as a control. Samples were processed for FRET measurements as described above.

*Cell-density-dependent signal processing:* The FRET reporter was induced as described above, and cells were grown to an $OD_{600\,nm}$ ~ 1.6. Before stimulation, the density of the reporter cells was adjusted by diluting or concentrating cells in an appropriate volume of cell-free supernatant, respectively. Stimulation then proceeded as described above.

**FRET acceptor-photobleaching experiments.** *Microscopy:* Experiments were performed on an Olympus IX81 inverted fluorescence microscope equipped with a 60× UPlanFLN 0.9 NA objective, a photomultiplier tube (Hamamatsu Photon Counting Head H7421-40, Hamamatsu City, Japan) and a 100 mW 515 nm laser (Cobolt, Sweden) that was coupled into the system via an AHF F73-014 z514 DCRB notch filter (Supplementary Fig. 2a). Data acquisition from the PMTs was performed as described by Sourjik et al.[35]. Fluorescence was excited with an MT20 illumination system. In order to attenuate CFP and minimize bleaching, the internal neutral density (ND) filter of the MT20 was set to 7.72%, and further reduced by an external (ND = 2) filter. A dense multilayer of cells was illuminated with excitation light centered around the CFP excitation maximum (EX: 438/24 nm, Dual BS 440/520 nm, EM: 475/23 nm) for the entire experiment to achieve continuous bleaching of CFP and avoid recovery effects. Prior to acceptor pho-tobleaching, CFP emission signals were measured for 60 s. The sample was then defocused by −16 μm to increase the bleaching area of the laser. The acceptor was bleached with the laser at maximum power for 20 s. After refocusing the sample, CFP emission was recorded for another 60 s. The efficiency of acceptor photo-bleaching was monitored by recording YFP fluorescence (EX: 504/12 nm, Dual BS 440/520 nm EM: 542/27 nm) for 6 s before and after each experiment. Further-more, after bleaching, a YFP image (EX: 504/12 nm, Dual BS 440/520 nm EM: 542/27 nm, 100% illumination intensity, 3 s exposure) was taken with a EMCCD Hamamatsu C9100-2 camera to check for homogenous bleaching of the sample area.

**Quantification of FRET.** The FRET efficiency was determined using the formula:

$$\text{FRET} = \frac{\text{CFP}_{post} - \text{CFP}_{pre}}{\text{CFP}_{post}} \cdot 100\%, \qquad (5)$$

where $\text{CFP}_{pre}$ and $\text{CFP}_{post}$ denote the emission levels before and after acceptor photobleaching, respectively. Since CFP is continuously excited, CFP will also bleach during periods of acceptor photobleaching. We thus correct for donor photobleaching by performing a linear fit to the CFP trajectories prior to and and after bleaching using the *robustfit* function in Matlab 2017b. $\text{CFP}_{pre}$ is then eval-uated at the end of the bleaching period by extrapolating the linear fit accordingly (see Fig. 2b). Each measurement records on the average fluorescence from hun-dreds of cells. Individual data points record the mean FRET from two technical replicates evaluated on the same gel pad (Supplementary Fig. 2c, d). For each experiment, we analyzed at least three biological replicates $n_e ≥ 3$. Barplots report the corresponding means, and error bars depict the respective standard deviations.

**E-FRET microscopy.** E-FRET imaging experiments were performed on a Nikon ECLIPSE Ti2 inverted fluorescence microscope equipped with a 100 mW 532 nM laser and a 60× Plan Apo λ 1.4 NA objective. Fluorescence was excited with an X-Cite Exact Illuminator, and fluorescence emission was detected with an Andor DU-897 EMCCD camera. The exposure time was 250 ms, and the EM gain was set to 150 for all channels. The following filters (EX/EM) and beam splitters (BS) were used: EX 504/12 nm, BS 520 nm and EM 554/23 nm for YFP; EX 436/10 nm, BS 455 nm, and EM 480/40 nm for CFP; and EX 436/10 nm, BS 455 nm and EM 554/23 nm for FRET. If required, the acceptor was bleached with a 532 nm laser (70% power) for 2 s.

**Quantification of E-FRET.** The apparent FRET efficiency $E_{app}$ was calculated as described in Zal and Gascoigne[48]:

$$E_{app} = \frac{I_{DA} - aI_{AA} - dI_{DD}}{I_{DA} - aI_{AA} + (G-d)I_{DD}}. \qquad (6)$$

Here $I_{DA}$, $I_{DD}$, and $I_{AA}$ refer to the fluorescence intensities measured in the donor (CFP), FRET, and acceptor (YFP) channels. Image registration of different channels was performed using the ImageJ plugins StackReg and MultiStackReg[68]. Image segmentation was performed using Ilastik 1.3.2[69]. Fluorescence

quantification was performed by ImageJ[67]. The measured fluorescence object intensities were corrected by subtracting the background signal and the cellular autofluorescence. The latter was determined by averaging the fluorescence intensities of nonfluorescent BIB1910 cells.

$a = I_{DA}(acc)/I_{AA}(acc)$ and $d = I_{DA}(don)/I_{AA}(don)$ correct for acceptor and donor bleed-through, respectively. $a$ and $d$ coefficients were determined from the fluorescence intensities from donor-only (CFP-RapA) and acceptor-only (Spo0F-YFP) samples from two biological replicates. In each case, data were acquired from 15 different fields of view. Further correction parameters $b = I_{DD}(acc)/I_{AA}(acc)$ and $b = I_{AA}(don)/I_{dd}(don)$ were negligible in our setup.

G refers to the G-factor given by:

$$G = \frac{(I_{DA} - a\,I_{AA} - d\,I_{DD}) - (I_{DA}^{post} - a\,I_{AA}^{post} - d\,I_{DD}^{post})}{I_{DD}^{post} - I_{DD}}, \quad (7)$$

where $I_{xx}^{post}$ refers to the fluorescence intensities after bleaching. The G-factor calibration was performed on the unstimulated FRET sample (RapA-CFP Spo0F-YFP) by acquiring images in each of the three fluorescence channels before and after acceptor photobleaching.

**Statistical analysis.** Matlab 2017b was used to determine the statistical significance of observed differences. Where applicable, unpaired $t$ test was used or one-way ANOVA with effect sizes given by Hedges' $g$ and $\eta^2$, respectively. The number of asterisks indicates the $P$-value with n.s. (nonsignificant): $P > 0.05$, *$P < 0.05$, **$P < 0.01$, ***$P < 0.001$. All $P$-values and relevant statistical parameters are provided in the Supplementary Data 1.

**Model of the FRET response.** Model: To describe how a population of cell processes an extracellular PhrA stimulus, we consider a population of identical (nonproducing) cells that are homogenously distributed in a volume $V_e$. The extra- and intracellular dynamics of the PhrA signal are described with the pump–probe model. The intracellular signal concentration $C_i(t) = C_i^s(t)$ follows from solving Eqs. (1)–(3) under the following conditions: initially, there is no signal inside the cell, the extracellular concentration is given by the stimulus $C_e^s(t=0) = C_{stim}$, and competing peptides are present at concentration $C_e^o$, if applicable. If not otherwise indicated, one can neglect cell growth and peptide degradation on the timescale of a typical stimulation experiment, i.e., $\mu \sim 0$, $\lambda_e \sim 0$, $\lambda_i \sim 0$ (see Supplementary Fig. 6; Fig. 3c). We assume that upon PhrA receptor binding, signal transduction to Spo0F occurs rapidly, i.e., FRET changes instantaneously as the intracellular PhrA levels vary with time, $FRET(t) = f(C_i(t))$. $f$ was modeled by:

$$FRET(t) = f(t) = FRET_0 - \Delta FRET \frac{C_i^n(t)}{EC_{50} + C_i^n(t)} \quad (8)$$

Here, $FRET_0$ is the FRET efficiency of unstimulated cells, $\Delta FRET$ the maximal response amplitude, $n$ the Hill coefficient, and $EC_{50}$ the intracellular peptide concentration that yields a half-maximal response.

*Parameter estimation:* This model has nine parameters in all, three of which were determined experimentally. First, the volume $V_i$ of a rod-shape bacteria was approximated by a cylinder with two semispherical caps, i.e., $V_i = \pi(L - D)\left(\frac{D}{2}\right)^2 + \frac{4}{3}\pi\left(\frac{D}{2}\right)^3$. Here, $L$ is the cell length, and $D$ is the cell diameter. Both were determined experimentally by measuring and averaging the lengths and widths of 150 reporter cells, which were imaged by bright-field microscopy using a 100×/1.4 NA objective. The extracellular volume was fixed at $V_e = 500\,\mu l$, and the number of cells $N_c$ was determined by cell counting using a C-Chip (Merck, Darmstadt). The growth rate μ was determined by separate fitting of the OD curve (inset Fig. 3c). The calculated $N_c$ value equivalent to an $OD_{600\,nm} = 1.6$ was $9.5 \times 10^7$.

All other parameters were estimated from parameter fitting. Fitting was performed using Matlab R2017b (MathWorks Inc.). The model equations were solved numerically using the *ode15s* solver. Parameters (with the exception of the intracellular peptide-degradation rate $\lambda_i$) were globally optimized by minimizing the sum of squared residuals (SSR) of all data sets with the *fsolve* function and the Levenberg–Marquardt algorithm. $\lambda_i$ was subsequently determined by fitting the FRET recovery experiment (Fig. 3c). The 95% confidence intervals for each parameter $\theta$ were determined from the following nonlinear constraint[70,71]:

$$\frac{SSR(\theta) - SSR\left(\hat{\theta}\right)}{SSR\left(\hat{\theta}\right)} \leq \frac{p}{n-p} F_{p,n-p}^\alpha. \quad (9)$$

where $p$ is the number of parameters, $n$ the number of data points, and $F^\alpha$ the value of the F-distribution for the α confidence level. Each parameter was minimized or maximized using the *fmincon* function with the above inequality as a nonlinear constraint, using interior-point optimization. The confidence intervals of the fitted curves were determined by bootstrapping of $10^4$ data sets that were randomly sampled from the data points. For each data set, we determined the best fit as described above, and determined the 95% confidence intervals from the 0.025 and 0.975 quantiles of all fits for each experimental condition.

## Data availability

All relevant data supporting the findings of the study are available in this article and its Supplementary Information files. The source data underlying Figs. 3c, Fig. 4b, c, Fig. 5, Fig. 6b, Fig. 7, Supplementary Fig. 1, Supplementary Fig. 2b–d, Supplementary Figs. 3–7, and Supplementary Fig. 8c are provided as a Source Data File.

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

## Acknowledgements

We thank B. Drees, B. Steinfeld, and T. Höfer for discussions, and C. Kaspar and K. Nagler for critical reading of the paper. We thank O. Kuipers, D. Rudner, D. Lopez, and A. Eldar for sending strains and plasmids, and we acknowledge instrument support by the Imaging Facility of the ZMBH. This work was funded by the Deutsche Forschungsgemeinschaft by an Emmy Noether Grant (BI1213/3–1) and a ERC Starting Grant (GA 260860) from the European Research Council awarded to I.B. I.B. is a member of CellNetworks. V.S. and I.B. also acknowledge support from the DFG Priority Program SPP1617 and the Max Planck Society.

## Author contributions

Conceived the project: I.B. Designed research: I.B., H.B., and V.S. Constructed strains: H.B., P.N., S.T., and S.S. Developed new experimental tools: H.B. and G.M. Performed experiments: H.B. and P.N. Developed the models: H.B. and I.B. Developed data analysis software: H.B. Performed the theoretical analysis: H.B., P.N., and I.B. Performed the statistical analysis: P.N. Supervised research: I.B. and V.S. Wrote the paper: I.B. with contributions from all authors.

## Competing interests

The authors declare no competing interests.
