## [Peer Review File · Nature Communications]

Reviewers' comments:

Reviewer #1 (Remarks to the Author):

This outstanding article addresses the question whether bacterial intercellular signaling systems utilizing peptide translocation pumps and cytoplasmic signal receptors (described as a 'pump and probe' scheme, as seen in RRNPP systems) provide advanced sensory capabilities beyond basic cell-density responses that characterize classic quorum sensing paradigms. In my opinion, this is a fundamental question in addressing why 'pump-probe' schemes evolved differently from most other bacterial intercellular signaling systems--what benefits do they provide? The objective here was to develop theoretical models that discern between different response patterns of peptide signaling, which include classic quorum-sensing, as well as chronometric or ratiometric abilities. To test the models, the authors constructed a FRET-based reporter system that fulfills the need to directly monitor ligand-receptor interactions within bacterial cells. By this assay, the authors deduced rates of peptide-signal transport into the cell and infer signal-transduction output responses in short (nearly immediate) time scales. The findings presented for the RapA-PhrA system are very convincing (and sufficient statistical analyses are provided). It would have been fortunate to demonstrate the full capacity of the experimental system by quickly depleting internal peptide levels and showing restoration of FRET signals on a short time scale. Unsuccessful attempts were made to address this (by testing pepF mutants) and, instead, the use of the scrambled PhrA and other Phr paralogs were sufficiently powerful substrates that demonstrate system specificity, if not reversibility. Three of the most important findings that stand out: the RapA-PhrA system integrates signal fluctuations over long time periods; competition for Opp transport between signal and non-signal peptides are transient; and, ratiometric information can be perceived by pump-probe systems.

Particularly enjoyable characteristics of this manuscript are the clear and concise writing qualities and the especially clean and intuitive figure designs (this is a great skill!).

Suggestions:

1. As the premise of this paper is based on unique attributes of pump-probe systems, it should be discussed whether chronometric or ratiometric sensing is possible for surface-sensing systems.

2. Description of the dose-dependence mode (page 6 of pdf file) is somewhat difficult to understand, particularly the phrase, "...the intracellular signal concentration approaches [equation]..." For many of us that are less inclined to understand such mathematical terms, could it be articulated another way as to what the concentration is approaching? The same is true for descriptions of ratiometric control later in the text (pg.7), where equation notation substitutes for a conceptual description.

3. In the section titled "Pump-probe networks..." (pg.7), where chronometric control is being described, shouldn't peptide degradation also be included as an important factor determining the rate of signal accumulation?

4. How are cells determined not be growing (pg. 9, fig3C)? Was an antibiotic added? I may have missed it in the methods, but it might be beneficial to mention in the results.

5. In Discussion, first paragraph (pg. 15), in reference to signal peptidases. Since Signal Peptidases are specific enzymes that process secretion signals from translocated proteins, should a different term be used that refers to peptidases that degrade peptide signals like PhrA?

Minor:

1. Intro, 1st paragraph (Pg.3). Perhaps change, "may not correlate at all well with the concentration..." to "may not correlate at all with the concentration..." OR "may not correlate well

with the concentration...”

2. Intro, last sentence of 1st paragraph (Pg.3). The first use of ‘pump-probe’ network only describes the ‘pump’ part adequately; perhaps a better description here what is meant by ‘probe’ (this is done well in the Results section, but it would help when the reader first encounters it. Is ‘pump-probe’ a new term?
3. Fig.1. it wasn’t immediately clear that the gray pac-man-like symbols represent peptide degradation (could a scissors symbol convey this more clearly?)
4. Fig. 1B, in the quorum sensing panel, switch the location of high density and low density graphic schemes (it makes more sense to have low density on the left, as one would relate low density on the left side of the graph axis).
5. (Pg.11) In pointing out agreement of data sets to fitted model, indicate lines in Fig 3B and 4D.
6. (Pg. 13) Is the inferred EC50 of 40+/-10uM based on table S1? (if so, please refer to table S1).
7. Figure 5 legend: Descriptions of A and B are switched compared to the figure.
8. Please cite reference 39 for SAMPs (pg.15).

Reviewer 2:

Summary of the manuscript

This manuscript combines novel experiments and theoretical analysis of the RapA-PhrA signaling system of *B. subtilis*, which serves as a model of the pump-probe RNPP systems found in many Gram positive bacteria. From the experimental perspective, the MS introduces a detailed set of measurements of the uptake dynamics of the PhrA peptide and its interaction with the RapA receptor, through the utilization of a FRET system which enables direct observation of RapA interaction with its target SpoOF. This allows the authors to study the impact of PhrA import and accumulation dynamics on a time scale of minutes. This precise measurement stands in contrast with previous indirect measurements of this system based on gene expression or phenotypic effects, which occur on a much longer time-scale and involve many additional factors.

The authors experimentally demonstrate several important observations:

1. PhrA is imported through the Opp system with rapid dynamics and high affinity. In a manner of minutes, externally provided phrA is cleared from the extracellular medium and is concentrated into the cells, leading to a reduction in RapA-SpoOF FRET activity.
2. Once phrA is internalized and FRET levels are reduced, they do not increase for a time-scale of hours when the cells do not grow, while FRET level do increase upon growth. This is interpreted to infer that PhrA is stable within the cell.
3. The two processes together imply that an external level of signal will be divided equally between all cells, leading to a concentration within the cells which is amplified by the small fractional volume employed by the cells and is proportional to the dose – the amount of externally supplied concentration divided by the number of cells.

The authors then use a mathematical model to analyze the importance of these processes for the function of pump-probe systems. They claim that the long integration time of the accumulation of the intracellular signal prevents the accurate utilization of this system as a cell-density measurement. In contrast, they claim that it fits very well with a ratiometric sensing system, where signaling level is proportional to the frequency of signal producers in the population.

Review

General

I think this is a very good paper which introduces a novel and highly appropriate technique to accurately measure signal dynamics within the cell. Nevertheless, I have multiple concerns about the experimental data and its interpretation which should be resolved before being able to accept this work. The theoretical model suggested by the authors is also incomplete and some of its implications are not stressed sufficiently, and seem to be rather strange. I recommend accepting the MS if the authors would be able to properly address these concerns and explain the model better.

Major concerns

SpoOF phosphorylation state: there is a whole level of complexity of the interaction of RapA and SpoOF which is completely ignored in the analysis of the data and may have a significant effect on it. RapA is a

phosphatase of Spo0F~P. The Perego lab (Ishikawa et al, 2002) has shown that Rap binds stably only to Spo0F~P and upon dephosphorylation, the complex becomes unstable and dissolves. There is very little direct binding of RapA to unphosphorylated Spo0F. This indicates that FRET levels should also depend on the phosphate state of Spo0F and introduces several problems to the interpretation of the results in general and the conclusion that PhrA is stable, specifically:

- 1. Level of phosphorylation at the used assay.** The FRET assay uses cells which are grown in minimal medium to OD~1. Bacillus cells often do not go into sporulation in minimal medium with high glucose level. It is therefore unclear to me if there any phosphorylation of Spo0F under these conditions at all? The level of Spo0F phosphorylation would depend on the expression and activity of the various kinases of the system, as well as on the levels of the other phosphorelay components, Spo0B and Spo0A. As the levels of all of these elements are unclear (and may vary with cell state), the interpretation of the results becomes more complex. Specifically, what is the level of kinase and phosphorelay activity in the conditions used by the authors? Minimal medium with high glucose, typically do not promote sporulation, so I'm not sure whether kinases are active at all in this state. Would results be different in a mutant of one of the kinases, or a *spo0B* mutant? Are the authors observing binding to Spo0F~P or the residual binding to Spo0F? What is the meaning of this complexity with respect to the maximal and minimal level of FRET activity observed? How would data analysis and interpretation depend on this additional layer of complexity?
- 2. Complex dissociation during experimental procedure:** If I understand correctly from the methods, there is a long delay between the time cells are washed (to eliminate extracellular PhrA levels) and FRET is measured under the microscope. Cells need to be washed, put on an agar piece, dry and be mounted under the microscope before a proper experiment can be done. This is ample time for Spo0F~P dephosphorylation to occur and complex to dissociate. Have the authors characterized the effect of this step on their results? How much error does it introduce to the measurements? I would note here also that it is not clear what is the exact procedure for changing the time of measurement (as in Fig. 3B) – Do they do it by changing the 5 minutes incubation in room temp, mentioned in the methods, to the specific duration?
- 3. Regain of FRET activity and the stability of phrA:** The authors do not see a recovery of FRET activity for a very long time when cells are kept non-growing (by the way, why are they not growing?), while seeing a regain in FRET levels in growing cells. They interpret this to indicate the lack of PhrA degradation and the effect of dilution due to growth. I can see two (non-exclusive) alternative mechanisms that may also explain these results:
 - a.** The lack of regain of activity may be the result of lack of kinase activity and not lack of degradation of the peptide. As cells revert to growth, kinase activity increases, leading to re-formation of the complex.
 - b.** Alternatively, PhrA may form a stable complex with RapA, which will protect it from degradation and prevent Rap rebinding to Spo0F~P. In this case, growth lead to the expression of additional Rap proteins which re-form the complex with Spo0F~P.

According to the dilution argument, an external level of 100nM instead of 10nM, should not recover at all, as even after ~3 cell cycles of dilution, the level should be too high. I think it would be illuminating to demonstrate this experimentally.

I think that addressing the above points in the data analysis and separating the different possibilities experimentally is crucial for proper analysis of the system. It may be that the authors have taken these problems into consideration and I'm missing a crucial point – I would love to be corrected if so.

PhrA uptake by OppA: The observed kinetic parameters for PhrA uptake by Opp are probably not significantly affected by the above complications (but maybe the apparent affinity is affected?). Yet, I have several additional questions and remarks on it:

1. The rate reported in this work is about 10-20 times higher than that reported by Lazzazera and Grossman (Cell, 1997) for PhrC. This should be mentioned and discussed.
2. Affinity: previous works on Opp and related pumps in other bacteria identified effective affinities on the order of 10 μ M-1mM (Fang, 2000 for dipeptide transport, Detmers, 1998 for *L. lactis* Opp). Even the smallest affinity is a 100 fold higher than the one mentioned here. Does this make sense, given the broad specificity of the *B. subtilis* Opp?

Peptide degradation and its relevant enzymes:

1. **PepF:** The authors find that despite a previous data on the impact of *pepF* overexpression on *phrA* levels, they see little effect of *pepF* deletion on their results. They suggest that *phrA* is not degraded as it is too short for PepF activity. Looking on the best characterized bacterium in this regard, *L. lactis*, it seems that this claim may very well be true. PepF in *L. lactis* is mostly involved in the cleavage of longer peptides. Shorter peptides like *phrA* are typically degraded in *L. lactis* by aminopeptidases, of which there are several in *B. subtilis*.
2. **Peptides as nutrients:** More generally, one must remember that the *B. subtilis* Opp system functions also for the uptake of short peptides as nutrients, and therefore there **has** to be a relevant degradation pathway for pentapeptides.
3. **Effect on Sporulation:** The author claim that no other peptidase has been identified as affecting sporulation. In fact, a mutation of the peptidase gene *papA* (*yqhT*, homologous to *Lactococcus pepP*) has a strong reduction in sporulation (which is the opposite phenotype of what would have happened for a *phrA* peptidase, but maybe the enzyme has pleiotropic effects). There is also another paralog named *papB*.

Dose dependence

Figure 5C shows that the kinetic behavior of the FRET reporter depends only on the dose (concentration/cell density) and not on the concentration and cell density independently. Given the authors' model, this should be true for the final FRET activity, but should not be the case for the dynamics. A lower cell density should increase the time it takes to approach the final activity. I think this is worth while demonstrating by repeating the experiment in Fig. 5C for the same doses but with a much lower cell-density (OD of ~0.1-0.2) – the final FRET activity should be the same but approach will be much longer – around 20 minutes to reach the level instead of <5 minutes. If this is impossible (due to the above kinase activity issues), then this point should at least be stressed when discussing the matter.

Dynamics at 100nM

The extracellular concentration dynamics described in Fig. 4D for extracellular level of 100nM is barely on par with the 95% confidence interval of predictions given by the 10nM measurements. What does this

most likely mean for the estimated parameters given both set of measurements? I think it would also help to clarify better the figure and caption.

Theoretical modeling and the impact of PhrA intracellular accumulation

The authors discuss the implications of their findings for the function of RNPP-type (Pump-probe) quorum-sensing systems. They claim that the long integration time does not fit well with a quorum-sensing function, but that with the fast uptake rate and long intracellular integration it fits well a ratiometric model.

I generally agree with the authors that a ratiometric function is highly likely, but I think that the authors fail to point some important implications of this model, by not considering a full model of the system, which includes also production and not only uptake. Here are my main concerns:

1. Pump-probe systems lead to a density independent (but ratiometric) signaling if one considers a full model, irrespective of phrA intracellular stability.

If we assume that the extracellular molecules are produced by a fraction f of the population with a production rate p and rapidly degraded by the cells, then the full equation for the extracellular concentration of phrA is:

$$\frac{dC_e^s}{dt} = \frac{N_c}{V_e} \times p \times f - \frac{N_c}{V_e} \times v_{mx} \frac{C_e^s}{K + C_e^s}$$

If we assume a constant number of cells and if $p < v_{mx}$ then this would lead to a steady state level of signal which is independent of the cell concentration but depends on the frequency of producers:

$$C_e^s(st) = K \frac{p \times f}{v_{mx} - p \times f}$$

This fits well with a ratiometric model. The time it will take to reach this steady-state depends on the uptake time scale and therefore on $\frac{N_c}{V_e}$, but for concentrations of $10^8 \frac{cells}{ml}$ used in this work, the uptake time-scale is <5 minutes. If there is no degradation of intracellular phrA, then the accumulation implies that intracellular phrA levels will be:

$$C_{int}(t) \sim p \times f \times t$$

The concentration will increase linearly with time. If PhrA is intracellularly degraded with a time-scale τ_{in} then the internal concentration would reach a steady-state:

$$C_{int}(st) \sim p \times f \times \tau_{in}$$

The ratiometric property is therefore independent of the fact that phrA is stable in the cell, but depends only on the fast uptake of PhrA from the medium.

2. Stability of PhrA is expected to overwhelm ratiometric measurement. I am not aware of reports on phrA extracellular levels during approach to sporulation, but Grossman and Lazzazera reported phrC levels of ~1 micromolar and Erez et al measured ~50-100nM for the aimP RNPP phage communication system. If one assumes that steady-state extracellular PhrA concentration is 50

nanomolar, then this implies (given the estimated uptake parameters) that $p \sim \frac{v_{mx}}{3} \sim 6 \times 10^4 \frac{\text{molecules}}{\text{min}} \sim 60 \frac{\mu\text{M}}{\text{min}}$. Under these conditions, the intracellular concentration of phrA would accumulate to inhibitory levels within ~40 seconds and would reach >mM levels in an hour. These levels are huge compared to the paper's predicted EC₅₀ of Phr on Rap-Spo0F~P binding (38 μM). Now, if the active population is just 10%, Phr would still accumulate to sufficient levels in ~5 minutes. If only 1% of the population produces Phr, it would still accumulate to sufficient levels to dissociate Rap-Spo0F~P in 50 minutes. Are these numbers reasonable given the long time-scale of the approach to sporulation? I am not sure.

One caveat to the above analysis is that Opp may be occupied by competing peptides during the approach to sporulation, which will reduce its effective import rate. Most of the competing peptides should have been eaten away by the cells by the time sporulation is initiated, though.

I expect the author to do the following given the above remarks:

1. Theory: Introduce and analyze a full model of the system – including both signal production, uptake and intracellular accumulation/degradation as well as additional factors such as competition from other peptides. They should discuss the possible implications of the full model for the ratiometric function and the quorum-sensing function. Given their estimated parameters, the impact of intracellular Phr accumulation to levels which far exceed the EC₅₀ levels should be discussed in the context of its effect on threshold cell densities and its implication for frequency measurements.
2. Experiments: Measure the effective concentration of phrA during approach to sporulation to be able to better estimate the relevant parameter regime. They can do that by using conditioned medium from sporulation in their FRET assay in a similar manner to what they have done in the paper to measure PhrA depletion from the conditioned medium.

Reviewer #3 (Remarks to the Author):

This is a well-written manuscript proposing that pump-probe networks can be modulated to perform different sensory tasks and mediate different regulatory functions, including quorum sensing, chronometric and ratiometric control. Authors describe the characteristics of the signal transduction system of prototypical PhrA-RapA system in *Bacillus subtilis*, using FRET to quantify extra- to intracellular signal conversion.

FRET results and methodology are fundamental for the understanding as well as significance of this paper. Therefore, this paper would be significantly improved by addressing the following issues relating to the FRET experiments:

- FRET experiments should include a single-donor as well as a single acceptor experiment to demonstrate lack of spectra-bleedthrough or FRET contamination. These controls would also indicate the background level of the FRET experiments indicating the meaning of "no FRET level".
- FRET microscopy is an imaging experiment. Images should be shown in supplementary material, at least for one relevant experiment.
- Role of acceptor concentration levels on FRET experiments need to be address to indicate the relevance of molecular crowding effect on FRET results.
- What parameter is being shown as FRET[-]? What does FRET[-] represent? FRET efficiency or some apparent relative FRET parameter that should then be clearly defined. How are these FRET values calculated? FRET efficiency, E%, or FRET ratios?
- Alternative FRET measurements using lifetime imaging or intensity based microscopy should be used to validate acceptor photobleaching.
- The FRET reporter is based on inter-molecular FRET so molecular crowding has to be addressed and excluded. Also bimolecular reporter should not be used to describe these FRET experiments since it may suggest fusion protein constructs.
- Authors describe FRET results as percent increase which do not allow for statistical evaluation and can be misleading. FRET results should be present more directly as fold increases/decreases and as much as possible absolute FRET efficiency numbers should be included allowing for readers to evaluate results.
- Statistical analysis should be provided and clearly described. It is not clear that actual statistical analysis has been performed to validate significance of results.

Response to Referees

We would like to thank the reviewers for dealing with our manuscript and for the very constructive comments provided. The revised manuscript has been modified in accordance with their comments and the editorial guidelines. In particular, we address the questions of Referee 3 in relation to the FRET methodology and have performed additional control experiments. We have also conducted additional experiments and data analysis using an extended pump-probe model to address the concerns of Referee 2 regarding the function of the FRET reporter, interpretation of FRET results and the ability of the PhrA signaling system to perform ratiometric sensing. Furthermore, we followed the suggestions of all referees, in particular of Referee 1, in order to improve the clarity of the manuscript and make it accessible for a broad readership, and we have streamlined the presentation to comply with the format requirements of the journal.

The most significant changes to the manuscript are summarized below:

- With the help of an extended theoretical model that explicitly models signal production and the effects of population growth, we demonstrate that pump-probe networks could in principle carry out a wide range of control functions (Fig. S1). This model allows for a more intuitive and less mathematical presentation of our simulation results.
- We improved our presentation of FRET methodology, which is based on quantification of FRET efficiency by means of fluorometric measurements of acceptor photobleaching (Fig. S3A), and validated the results using E-FRET imaging (Fig. S8).
- We have transferred data on the FRET negative control from the Supplement to Fig. 2C, and performed additional control experiments (Fig. S3B) which corroborate that the percentage increase in donor fluorescence from reporter cells indeed reflects on the FRET efficiency.
- We show that perturbations of phosphorelay signaling have no effect on FRET in unstimulated or stimulated cells, thus confirming that changes in FRET report specifically on effects of signaling via the PhrA-RapA-Spo0F pathway (Fig. S4B, Fig. 2C).
- We quantitatively fit the activation and long-term recovery response of *B. subtilis* to PhrA stimulation in growing populations to the pump-probe model, and find that dilution due to growth and intracellular peptide degradation contribute in roughly equal parts to the deactivation of the pathway (Fig. 3C).
- We perform model validation using the response dynamics to a weaker stimulus (Fig. 4D) and discuss minor deviations seen in experimental data for the stronger 100 nM stimulus (Fig. S6).
- We estimate the extracellular PhrA concentration in supernatants of wild-type populations using a sensitized bioassay (Fig. S7) in order to provide further evidence for the contention that the parameters of the network are properly balanced to facilitate ratio-sensing in heterogeneous populations.

These changes have been highlighted in the revised manuscript and the Supplementary Material.

Please find our point-by-point responses to the Referees' comments below. We hope that the revised manuscript is now acceptable for publication in Nature Communications

Reply to the comments of Referee 1

This outstanding article addresses the question whether bacterial intercellular signaling systems utilizing peptide translocation pumps and cytoplasmic signal receptors **(described as a ‘pump and probe’ scheme, as seen in RRNPP systems) provide** advanced sensory capabilities beyond basic cell-density responses that characterize classic quorum sensing paradigms. In my opinion, this is a fundamental question in **addressing why ‘pump-probe’ schemes evolved differently from most other bacterial** intercellular signaling systems--what benefits do they provide? The objective here was to develop theoretical models that discern between different response patterns of peptide signaling, which include classic quorum-sensing, as well as chronometric or ratiometric abilities. To test the models, the authors constructed a FRET-based reporter system that fulfills the need to directly monitor ligand-receptor interactions within bacterial cells. By this assay, the authors deduced rates of peptide-signal transport into the cell and infer signal-transduction output responses in short (nearly immediate) time scales. The findings presented for the RapA-PhrA system are very convincing (and sufficient statistical analyses are provided). It would have been fortunate to demonstrate the full capacity of the experimental system by quickly depleting internal peptide levels and showing restoration of FRET signals on a short time scale. Unsuccessful attempts were made to address this (by testing pepF mutants) and, instead, the use of the scrambled PhrA and other Phr paralogs were sufficiently powerful substrates that demonstrate system specificity, if not reversibility. Three of the most important findings that stand out: the RapA-PhrA system integrates signal fluctuations over long time periods; competition for Opp transport between signal and non-signal peptides are transient; and, ratiometric information can be perceived by pump-probe systems. Particularly enjoyable characteristics of this manuscript are the clear and concise writing qualities and the especially clean and intuitive figure designs (this is a great skill!).

We thank the reviewer very much for this highly appreciative appraisal of our work, for the very careful review and the constructive criticism and excellent suggestions provided, which have greatly helped us to improve the manuscript further. Please see our point-by-point reply below.

1. As the premise of this paper is based on unique attributes of pump-probe systems, it should be discussed whether chronometric or ratiometric sensing is possible for surface-sensing systems.

We agree. The capacity for ratiometric or chronometric response schemes is not unique to pump-probe systems. For example, the yeast pheromone system performs ratiometric sensing. However, this ability relies on a quite complex control mechanism that involves feedback-regulated pheromone degradation (Banderas et al., 2016). Similarly, it is conceivable that a surface-sensing system could implement chronometric control given suitable additional regulation. However, pump-probe systems achieve this functional versatility without additional regulation.

We have revised the corresponding paragraph in the Discussion (Lines 489f) to explicitly state that the yeast pheromone pathway uses membrane-based receptors and is nevertheless capable of ratio sensing, but requires specific regulatory features to do so.

Banderas A, Koltai M, Anders A, Sourjik V. Sensory input attenuation allows predictive sexual response in yeast. *Nat Commun.* Nature Publishing Group; 2016;7: 12590. doi:10.1038/ncomms12590

2. Description of the dose-dependence mode (page 6 of pdf file) is somewhat difficult to **understand, particularly the phrase, “...the intracellular signal concentration approaches [equation]...”** For many of us that are less inclined to understand such mathematical terms, could it be articulated another way as to what the concentration is approaching? The same is true for descriptions of ratiometric control later in the text (pg.7), where equation notation substitutes for a conceptual description.

We agree that a conceptual presentation of modeling results will help to improve the accessibility of the manuscript.

We have followed the suggestion of Referee 2 to explicitly model signal production (and we have also included a model for cell growth) in order to demonstrate that pump-probe networks could carry out various functions (Lines 135f). This allows for a more intuitive description of our results, and we have made an effort to minimize the use of mathematical terms (Lines 145f).

3. In the section titled “Pump-probe networks...” (pg.7), where chronometric control is being described, shouldn’t peptide degradation also be included as an important factor determining the rate of signal accumulation?

This is correct. To clarify the point, we have included two examples of simulation results, which show that varying the intracellular degradation rate can tune signal accumulation and thus the chronometric delay (Fig. S1C).

4. How are cells determined not be growing (pg. 9, fig3C)? Was an antibiotic added? I may have missed it in the methods, but it might be beneficial to mention in the results.

Thanks for spotting this. Indeed the methods used for the recovery phase were not well explained in the Methods and the Results sections. Non-growth conditions emerged “naturally” when cells were stimulated in microtubes and incubated at 37° C on a thermoshaker. For growth, cells were stimulated in culture tubes and grown at 37° C on a regular shaker. This is now explicitly stated in the Results Section (Lines 238-240). We have also added a paragraph on FRET recovery response (Lines 608f) to the Methods. Please note that, in response to comments made by Referee 2, we can now account for the recovery response of growing cells within the first hour with the help of the pump-probe model. We have therefore transferred the data on non-growing cells to the Supplementary Material Fig. S5A and rewritten the corresponding paragraph in the Results.

5. In Discussion, first paragraph (pg. 15), in reference to signal peptidases. Since Signal Peptidases are specific enzymes that process secretion signals from translocated proteins, should a different term be used that refers to peptidases that degrade peptide signals like PhrA?

We agree that the use of the term signal peptidase is misleading. We have removed the term from the manuscript.

Minor

1. Intro, 1st paragraph (Pg.3). Perhaps change, “may not correlate at all well with the concentration...” to “may not correlate at all with the concentration...” OR “may not correlate well with the concentration...”

Thank you. We changed the text to “signal concentration may not correlate” (Line 74).

2. Intro, last sentence of 1st paragraph (Pg.3). The first use of ‘pump-probe’ network only describes the ‘pump’ part adequately; perhaps a better description here what is meant by ‘probe’ (this is done well in the Results section, but it would help when the reader first encounters it. Is ‘pump-probe’ a new term?

We agree. The term “pump-probe” is a new term, which succinctly describes the key features of the signaling architecture. We moved the description of the probe component of this concept to the Introduction (Line 73f).

3. Fig.1. it wasn’t immediately clear that the gray pac-man-like symbols represent peptide degradation (could a scissors symbol convey this more clearly?)

We apologize for confusing the referee. We have replaced the pac-man symbols by a scissors, and explain the meaning of the symbol in the caption to Fig. 1A .

4. Fig. 1B, in the quorum sensing panel, switch the location of high density and low density graphic schemes (it makes more sense to have low density on the left, as one would relate low density on the left side of the graph axis).
We agree, and have swapped the corresponding graphs in Fig. 1B.

5. (Pg.11) In pointing out agreement of data sets to fitted model, indicate lines in Fig 3B and 4D.
We agree. The reference is made in Line 306.

6. (Pg. 13) Is the inferred EC50 of 40+/-10uM based on table S1? (if so, please refer to table S1).
This is correct. In fact, all inferences in this section derive from the parameters given in Table 1. The reference to Table 1 is now made explicit at the beginning of the corresponding paragraph (Line 321).

7. Figure 5 legend: Descriptions of A and B are switched compared to the figure.
Thanks for spotting this! We have replaced Fig. 5 by a revised figure and in which the top two panels are correctly referenced.

8. Please cite reference 39 for SAMPs (pg.15).
We agree. The reference is now cited in Line 425.

Reply to the comments of Referee 2

Summary of the manuscript

This manuscript combines novel experiments and theoretical analysis of the RapA-PhrA signaling system of *B. subtilis*, which serves as a model of the pump-probe RNPP systems found in many Gram positive bacteria. From the experimental perspective, the MS introduces a detailed set of measurements of the uptake dynamics of the PhrA peptide and its interaction with the RapA receptor, through the utilization of a FRET system which enables direct observation of RapA interaction with its target Spo0F. This allows the authors to study the impact of PhrA import and accumulation dynamics on a time scale of minutes. This precise measurement stands in contrast with previous indirect measurements of this system based on gene expression or phenotypic effects, which occur on a much longer time-scale and involve many additional factors.

The authors experimentally demonstrate several important observations:

1. PhrA is imported through the Opp system with rapid dynamics and high affinity. In a manner of minutes, externally provided phrA is cleared from the extracellular medium and is concentrated into the cells, leading to a reduction in RapA-Spo0F FRET activity.
2. Once phrA is internalized and FRET levels are reduced, they do not increase for a time-scale of hours when the cells do not grow, while FRET level do increase upon growth. This is interpreted to infer that PhrA is stable within the cell.
3. The two processes together imply that an external level of signal will be divided equally between all cells, leading to a concentration within the cells which is amplified by the small fractional volume employed by the cells and is proportional to the dose – the amount of externally supplied concentration divided by the number of cells.

The authors then use a mathematical model to analyze the importance of these processes for the function of pump-probe systems. They claim that the long integration time of the accumulation of the intracellular signal prevents the accurate utilization of this system as a cell-density measurement. In contrast, they claim that it fits very well with a ratiometric sensing system, where signaling level is proportional to the frequency of signal producers in the population.

Review

General

I think this is a very good paper which introduces a novel and highly appropriate technique to accurately measure signal dynamics within the cell. Nevertheless, I have multiple concerns about the experimental data and its interpretation which should be resolved before being able to accept this work. The theoretical model suggested by the authors is also incomplete and some of its implications are not stressed sufficiently, and seem to be rather strange. I recommend accepting the MS if the authors would be able to properly address these concerns and explain the model better.

We thank the referee both for the positive remarks on our work, and for the many insightful and constructive comments and excellent suggestions. These have been very helpful in improving the manuscript. In the revised manuscript we address the concerns regarding the interpretation of experimental FRET data by additional experiments addressing phosphorylation of Spo0F- via the sporulation phosphorelay to confirm that our study addresses signal processing by the PhrA-RapA-Spo0F pathway. Furthermore, we have extended the theoretical model to include the effects of dilution due to cell growth, and performed a model-based analysis of the FRET recovery response which suggests that PhrA is degraded at a rate comparable to the cellular growth rate under our experimental conditions. We also include additional data and analysis that provide further evidence for the contention that the PhrA signaling system is capable of ratiosensing given the inferred parameters. Finally, we have followed the referee's suggestion to include signal production in order to explain that pump-probe networks could in principle mediate various control functions. Please find our point-by-point response below.

Major concerns

C1. Spo0F phosphorylation state: there is a whole level of complexity of the interaction of RapA and Spo0F which is completely ignored in the analysis of the data and may have a significant effect on it. RapA is a phosphatase of Spo0F~P. The Perego lab (Ishikawa et al, 2002) has shown that Rap binds stably only to Spo0F~P and upon dephosphorylation, the complex becomes unstable and dissolves. There is very little direct binding of RapA to unphosphorylated Spo0F. This indicates that FRET levels should also depend on the phosphate state of Spo0F and introduces several problems to the interpretation of the results in general and the conclusion that PhrA is stable, specifically:

The referee points out that signaling via the sporulation phosphorelay could introduce an additional level of complexity. We agree: PhrA is processed via two interconnected pathways - the PhrA-RapA-Spo0F and the PhrA-RapA-Spo0F~P.

In this study, we chose to focus on PhrA signal processing under conditions in which PhrA signaling is active in wild-type cells, while at the same time avoiding the complexities of stress response signaling by Spo0F~P. This is why we cultivated cells in S7 media. We thus investigate signaling by the RapA-Spo0F pathway (see also C2 below). We apologize that this point was not clearly communicated in the previous version of our manuscript.

In order to clarify this point, we now explicitly state that signaling by the phosphorelay could introduce an additional level of complexity (line 218ff). However, under our experimental conditions, FRET reports specifically on PhrA signal processing via the RapA-Spo0F-pathway. We support this statement with new data on a phosphorelay triple mutant kinA kinB spo0B (Fig. 2C) and additional data, including single and double mutants, in the Supplementary Material Fig. S4B.

C2. Level of phosphorylation at the used assay. The FRET assay uses cells which are grown in minimal medium to OD~1. Bacillus cells often do not go into sporulation in minimal medium with high glucose level. It is therefore unclear to me if there any phosphorylation of Spo0F under these conditions at all? The level of Spo0F phosphorylation would depend on the expression and activity of the various kinases of the system, as well as on the levels of the other phosphorelay components, Spo0B and Spo0A. As the levels of all of these elements are unclear (and may vary with cell state), the interpretation of the results becomes more complex. Specifically, what is the level of kinase and phosphorelay activity in the conditions used by the authors? Minimal medium with high **glucose, typically do not promote sporulation, so I'm not sure** whether kinases are active at all in this state.

The referee asks us to more clearly define our experimental conditions with respect to effects of phosphosignaling on FRET: Phosphorelay activity is low and thus it does not influence FRET (see below).

Would results be different in a mutant of one of the kinases, or a spo0B mutant?

No. To confirm that our assay specifically reports on effects of PhrA on the RapA-Spo0F pathway we deleted kinA, kinB, both kinases together, kinA kinB and spo0B, respectively as requested. FRET remained unchanged under all conditions for both unstimulated and stimulated cells, as expected.

The resulting data is now included in Fig. 2C and Fig. S4B, respectively.

Are the authors observing binding to Spo0F~P or the residual binding to Spo0F?

Under our experimental conditions, FRET is dominated by interactions in the RapA - Spo0F pathway. See above.

What is the meaning of this complexity with respect to the maximal and minimal level of FRET activity observed? How would data analysis and interpretation depend on this additional layer of complexity?

Under our experimental conditions, changes in FRET report on effects caused by PhrA signaling via the RapA-Spo0F pathway. More complex signaling situations (involving Spo0F~P) required to tackle the questions raised by the referee will be addressed in future work.

To clarify the role of phosphorelay signaling we now explicitly state that the PhrA-RapA-Spo0F pathway is embedded in a complex signaling network and specifically mention phosphorelay signaling and other Rap-Spo0F pathways (line 218f). We have included data on the triple kinA kinB spo0B mutant Fig. 2C and 4SB) to support the assertion that changes in FRET result from altered activity in the RapA-Spo0F-pathway (line 224f).

C3. Complex dissociation during experimental procedure: If I understand correctly from the methods, there is a long delay between the time cells are washed (to eliminate extracellular PhrA levels) and FRET is measured under the microscope. Cells need to be washed, put on an agar piece, dry and be mounted under the microscope before a proper experiment can be done. This is ample time for Spo0F~P dephosphorylation to occur and complex to dissociate.

Under our experimental conditions, FRET reports on the activity of the PhrA-RapA-Spo0F pathway only; thus, effects of Spo0F dephosphorylation are negligible. As an additional control, we have applied both saturating (10 μ M) and sub-saturating (10 nM) PhrA stimuli to the various phosphorelay mutants kinA, kinB, and spo0B. There is no significant difference in the responses of mutant and WT reporter cells (Fig. S4B, second and third panels).

Have the authors characterized the effect of this step on their results?

How much error does it introduce to the measurements?

The referee is correct in stating that there is a delay between stimulation and the actual FRET measurement. This measurement delay has little effect on FRET.

First, we have always performed at least two FRET measurements on the same sample with varying delays between measurements, sometimes more than 40 minutes (Fig. S3C, previous Fig. S3B). To make this point explicit in the revised manuscript we have specifically marked the data-points in the scatter plot that were executed at times more than 40 minutes apart and included corresponding data on stimulated cells (Fig. S3D). In both cases, the overall biological variability is substantially larger than the variability from technical replicates with a measurement delay.

Second, the "FRET recovery experiment" under non-growth conditions shows that FRET does not change for 3 h after stimulation (previous Fig. 3C, Fig. S5A in the revised manuscript).

Together this data shows that FRET is quite robust to measurement delays and that our experimental procedure is sufficiently accurate to report on effects that actually result from parameter variations in signal stimulation.

I would note here also that it is not clear what is the exact procedure for changing the time of measurement (as in Fig. 3B). Do they do it by changing the 5 minutes incubation in room temp, mentioned in the methods, to the specific duration?

This is correct. We varied the incubation time. "Time" in Fig. 3B refers to the time spent by the cells in the medium after adding the stimulus (incubation time + 1 minute for centrifugation). To clarify, we now explicitly define t_s in Materials and Methods (line 585f) and have accordingly changed the x-axis label in all figures depicting data on signaling dynamics plotted as a function of t_s to Time t_s (Fig. 3B and C, Fig. 4D, Fig. 5A,B,C, Fig. S5, Fig. S6).

C4. Regain of FRET activity and the stability of phrA: The authors do not see a recovery of FRET activity for a very long time when cells are kept non-growing (by the way, why are they not growing?), while seeing a regain in FRET levels in growing cells.

Thanks for spotting this. Indeed the methods used for the recovery experiments were not well explained. Non-growth conditions emerged “naturally” when 500 μ l cell samples were incubated at 37° C on a thermoshaker. For growth, the stimulus was added directly to a 9 ml shake-flask culture. This is now explicitly stated in the Results Section (Line 238f). We have also introduced a separate paragraph on the FRET recovery response (Lines 608f) to the Methods to provide more experimental details.

They interpret this to indicate the lack of PhrA degradation and the effect of dilution due to growth. I can see two (non-exclusive) alternative mechanisms that may also explain these results:

We thank the referee very much for these interesting suggestions for alternative interpretations of our experiments, which have prompted further analysis of our data. As will be explained below in more detail, we succeeded in fitting the FRET recovery response under growing conditions within the framework of an extended pump-probe model which considers effects on the FRET response arising from both signal dilution due to growth and intracellular peptide degradation, and obtained an excellent fit to the data (Fig. 3C), indicating that, under growth conditions, PhrA is degraded at a rate which is comparable to that of cell growth. We have modified the Results section and the Discussion accordingly.

a. The lack of regain of activity may be the result of lack of kinase activity and not lack of degradation of the peptide. As cells revert to growth, kinase activity increases, leading to re-formation of the complex.

We can exclude the possibility that differential kinase activity is responsible for the observed differences, as FRET is not affected by perturbations to phospho-signaling under our conditions (see responses to C1 and Fig. S4B).

b. Alternatively, PhrA may form a stable complex with RapA, which will protect it from degradation and prevent Rap rebinding to Spo0F~P. In this case, growth lead to the expression of additional Rap proteins which re-form the complex with Spo0F~P.

It is possible that PhrA-RapA may protect PhrA from degradation. However, any loss of free PhrA due to degradation should affect complex formation (and thus FRET) in non-growing cells, provided the reaction kinetics at the receptor is fast relative to all other processes. The assumption of a fast reaction kinetics is a plausible one, given that the activation dynamics is limited by signal uptake and the relatively high $K_d \sim \mu$ M for Rap-Phr interactions observed for other Raps (Gallego del Sol, 2013). However, we agree that, in the absence of a direct experimental proof for a fast receptor kinetics, a caveat is required, which we we have now added to the Discussion (Line 400f).

F. Gallego del Sol and A. Marina, PLOS Biology 11(3), e1001511, 2013
<https://journals.plos.org/plosbiology/article?id=10.1371/journal.pbio.1001511>

According to the dilution argument, an external level of 100nM instead of 10nM, should not recover at all, as even after ~3 cell cycles of dilution, the level should be too high. I think it would be illuminating to demonstrate this experimentally.

The referee proposes an interesting experiment to clarify the underlying mechanism for recovery of FRET during growth and makes a specific suggestion to investigate this. However, as explained below, the proposed experiment will yield results that are inconclusive. We have now simulated the proposed experiment (supplying a 100 nM stimulus to an exponentially

growing population of cells and assuming that there is no signal degradation). In contrast to the referee's expectations, the model predicts that recovery of FRET already sets in within the first cell cycle. The experiment would therefore be inconclusive.

Very high signal concentrations are required to prevent recovery in theory (red curve); however, it is very likely that the model is invalid for describing actual system behavior. The required concentrations (10 μM) are several orders of magnitude higher than measured extracellular signal concentrations for PhrA (Fig. S7) or any other RRNPP-type signal peptide. Such high concentrations could give rise to additional effects (e.g. due to a non-linear concentration dependence of peptide degradation) that are irrelevant under physiological conditions. Second, signal uptake will take ~ 2 hours to complete. During this time, FRET rises in unstimulated cells (Fig. S5B, grey line), presumably due to ongoing production of Spo0F and RapA molecules. This will affect the dose-response curve and presumably shift the EC_{50} to higher PhrA levels (Babel et al., 2016). These complexities make data from a 10 μM recovery experiment inherently difficult to interpret.

Babel H, Bischofs IB. Molecular and Cellular Factors Control Signal Transduction via Switchable Allosteric Modulator Proteins (SAMPs). *BMC Syst Biol.* BioMed Central; 2016;10: 35. doi:10.1186/s12918-016-0274-3

The above analysis suggests that the most informative experiment is the recovery response to 10 nM PhrA presented in the original manuscript. Here, signal uptake is completed in less than 10 minutes. Moreover, RapA/Spo0F levels remain roughly constant in unstimulated controls for about an hour (Fig. S5B, grey line). We reasoned that one could infer contributions from dilution and degradation by fitting the data from the first hour of the recovery experiment to the extended pump-probe model, described above. Indeed, we obtain an excellent fit of the model to the data (Fig. 3C). We find that FRET recovers faster than predicted from growth-based dilution alone and the best fit implies an effective signal degradation rate of $\lambda_i = 0.63 \text{ h}^{-1}$, which is comparable to the growth-based dilution rate $\mu = 0.58 \text{ h}^{-1}$.

We have revised the manuscript accordingly: Fig. 3C focusses on the FRET recovery response in a growing population during the first hour (Line 240f). We expanded the pump-probe model to consider effects from signal dilution owing to cell growth (Line 664f, Line 701) and included the data to fit the intracellular degradation rate (Line 302f, Table 1). We describe the implications of peptide degradation for the signal processing time and the ability of the system to track and integrate over extracellular signals (Line 319f). Finally, to avoid overloading Fig. 3, the data on the deactivation response under non-growth conditions (Fig. S5A), and the extended time course of the FRET recovery response under growth conditions (Fig. S5B) was moved to the Supplement.

I think that addressing the above points in the data analysis and separating the different possibilities experimentally is crucial for proper analysis of the system. It may be that the authors have took these problems into consideration and I'm missing a crucial point – I would love to be corrected if so.

We thank the referee for the constructive criticism, which has helped to improve our manuscript. We believe that, with the data and clarifications provided in the revised manuscript, we have built a convincing foundation for the analysis of the system.

C5. PhrA uptake by OppA: The observed kinetic parameters for PhrA uptake by Opp are probably not significantly affected by the above complications (but maybe the apparent affinity is affected?). Yet, I have several additional questions and remarks on it:

1. The rate reported in this work is about 10-20 times higher than that reported by Lazzazera and Grossman (Cell, 1997) for PhrC. This should be mentioned and discussed.

The referee claims a discrepancy in our rate estimate with respect to previous data from radioactive labeling experiments by Lazazzera et al. However, as explained below, the rates are actually in quite good agreement with each other.

Lazazzera et al. estimated an initial uptake rate of $v \sim 2000$ molecules/min per cell at an extracellular PhrC concentration of 10nM. At the same extracellular PhrA concentration, our model predicts an uptake rate of $v \sim 12.000 \pm 3000$ molecules/min per cell. The agreement in the transport rate estimates derived from two datasets obtained using different methods, peptides and strains and cultivation conditions is actually quite remarkable since, in general, the rate of cellular peptide uptake is controlled by the expression levels of the Opp system. Rates could also vary depending on the nature of the imported peptide (PhrA vs PhrC).

We thus state in the Discussion that our rate estimates for Opp-based transport are in good agreement with previous work (Line 403f) and also mention that peptide import at physiological concentrations is much slower than v_{\max} (Line 409f).

B.A. Lazazzera, J.M. Solomon and A.D. Grossman, Cell (89), 917-925, 1997

2. Affinity: previous works on Opp and related pumps in other bacteria identified effective affinities on the order of 10 μ M-1mM (Fang, 2000 for dipeptide transport, Detmers, 1998 for *L. lactis* Opp). Even the smallest affinity is a 100 fold higher than the one mentioned here. Does this make sense, given the broad specificity of the *B. subtilis* Opp?

The referee wonders how the discrepancy in the apparent affinity of peptide transport in *Lactococcus lactis* and *B. subtilis* can be explained. In *L. lactis* cellular transport saturates in the upper μ M regime, while (based on our data) in *B. subtilis* transport saturates at lower concentrations, namely in the sub- μ M regime. This can be readily interpreted as indicating high-affinity transport of PhrA in *B. subtilis* and low-affinity transport of peptides in *L. lactis*.

The major determinant of the apparent affinity of peptide transport is the oligopeptide binding protein OppA. The *L. lactis* OppA exhibits very low sequence identity to *B. subtilis* OppA. It is an outlier in the OppA family and transports a much wider range of peptides than any other bacterial oligopeptidase known (Detmers, 1998. Detmers et al., 2000, Berntsson et al., 2009). *L. lactis* may have adapted its transport machinery for growth in peptide-rich environments such as milk, and may not utilize OppA for signaling purposes. In *Enterococcus faecalis*, signal transport by Opp uses a specific peptide-binding protein PgrZ that has high affinity for the signal. *B. subtilis* lacks specific peptide-binding proteins for Phr signals. Our data therefore indicates that the *B. subtilis* OppA has evolved a sufficiently high affinity to play a role in Phr signaling.

To clarify, we have expanded the discussion of K_m values for different Opp transporters in the context of low signal concentrations and competition for peptide (line 409f).

F. Detmers et al., *Biochemistry*, 37, 16671 – 16679, 1998
<https://www.rug.nl/research/portal/files/3618651/1998BiochemDetmers.pdf>

F. Detmers et al., *Proc. Natl. Acad. Sci USA* 97: 12487 – 12492, 2000
<https://www.pnas.org/content/pnas/97/23/12487.full.pdf>

R. Berntsson et al., *EMBO Journal* 28, 1332-1340, 2009
<https://www.embopress.org/doi/pdf/10.1038/emboj.2009.65>

C6. Peptide degradation and its relevant enzymes:

1. PepF: The authors find that despite a previous data on the impact of pepF overexpression on phrA levels, they see little effect of pepF deletion on their results. They suggest that phrA is not degraded as it is too short for PepF activity. Looking on the best characterized bacterium in this regard, *L. lactis*, it seems that this claim may very well be true. PepF in *L. lactis* is mostly involved in the cleavage of longer peptides. Shorter peptides like phrA are typically degraded in *L. lactis* by aminopeptidases, of which there are several in *B. subtilis*.

The referee points out that there might exist alternative pathways that could degrade PhrA in addition to PepF. We agree. As with cleavage of extracellular pre-peptides, intracellular peptide degradation could involve redundant enzymes. However, it is beyond the scope of the manuscript to identify additional peptidases. To clarify this, we mention that additional peptidases may contribute to degradation, when introducing the PepF data (Line 245f).

2. Peptides as nutrients: More generally, one must remember that the *B. subtilis* Opp system functions also for the uptake of short peptides as nutrients, and therefore there has to be a relevant degradation pathway for pentapeptides.

The referee points out that the Opp system also imports peptides that could serve as nutrients and thus there must be a pathway to degrade them. We agree. As explained in response to C4, the quantitative analysis of the FRET recovery experiment indicates that PhrA is degraded in the cell at a very slow rate (Fig. 3C, Table 1).

We would like to point out that (very) slow degradation of PhrA need not necessarily imply that cells could not utilize the same pathway for feeding on peptides. Nutrient peptides and signaling peptides likely accumulate to very different levels in the cell. Assuming that cells take up nutrient peptides at similar rates, overall substrate concentrations will reach mM concentrations easily. Hence, even an unspecific peptidase with a broad substrate spectrum and a low affinity for peptides could generate a sufficiently high flux of amino acids. In contrast, a specific peptide at physiological concentrations for intracellular signaling ($EC_{50} \sim \mu\text{M}$) will be degraded at a much slower rate and will compete with all other peptides for degradation. In this respect, it is noteworthy that our data (Fig. S5C) shows a small but significant effect of both a pepF deletion and PepF overexpression on FRET when cells were stimulated with 10 μM PhrA (an unphysiologically high amount of signal). This observation is now explicitly stated in the caption to Fig. S5C.

3. Effect on Sporulation: The author claim that no other peptidase has been identified as affecting sporulation. In fact, a mutation of the peptidase gene papA (yqhT, homologous to *Lactococcus* pepP) has a strong reduction in sporulation (which is the opposite phenotype of what would have happened for a phrA peptidase, but maybe the enzyme has pleiotropic effects). There is also another paralog named papB.

We thank the referee for pointing out PapA as an alternative peptidase. We agree its sporulation phenotype is not consistent with a role in PhrA signaling. To the best of our

knowledge, no peptidase is known to act specifically on PhrA or on any other Phr signal in *B. subtilis*. Future work is required to address whether other peptidases contribute to the degradation of PhrA.

C7. Dose dependence

Figure 5C shows that the kinetic behavior of the FRET reporter depends only on the dose (concentration/cell density) and not on the concentration and cell density **independently. Given the authors' model, this should be true for the final FRET activity**, but should not be the case for the dynamics. A lower cell density should increase the time it takes to approach the final activity. I think this is worth while demonstrating by repeating the experiment in Fig. 5C for the same doses but with a much lower cell-density (OD of ~0.1-0.2) – the final FRET activity should be the same but approach will be much longer – around 20 minutes to reach the level instead of <5 minutes. If this is impossible (due to the above kinase activity issues), then this point should at least be stressed when discussing the matter.

We agree. The kinetics is predicted to be sensitive to concentration, while the final output only depends on the dose. This is now stated explicitly in the text (Line 348f).

Dynamics at 100nM

The extracellular concentration dynamics described in Fig. 4D for extracellular level of 100nM is barely on par with the 95% confidence interval of predictions given by the 10nM measurements. What does this most likely mean for the estimated parameters given both set of measurements? I think it would also help to clarify better the figure and caption.

We performed a refitting of the data as requested. When both datasets were included for fitting, the estimates of most parameters were unaffected (less than 25% change) except for the uptake parameters which changed by -28% (v_{max}) and +49% (K_m). However, while this improved the fit to the 100nM (Fig. S6B), the 10nM data was captured less well (not shown). The original parameters were then used to predict the response to a 30nM stimulus and the experimental data falls within the confidence interval. For the revised manuscript, we thus included the 30nM prediction in the Results (Fig. 4C) and moved the 100 nM data (prediction (Fig. S6A) and re-fit (Fig. S6B) to the Supplement. In the Discussion (Line 396f), we add a caveat to the effect that at high signal concentration additional effects may come into play, although it is unclear whether they have any biological relevance, given that physiological signal concentrations are much lower (see Fig. S7)

C8. Theoretical modeling and the impact of PhrA intracellular accumulation

The authors discuss the implications of their findings for the function of RNPP-type (Pump-probe) quorum sensing systems. They claim that the long integration time does not fit well with a quorum-sensing function, but that with the fast uptake rate and long intracellular integration it fits well a ratiometric model. I generally agree with the authors that a ratiometric function is highly likely, but I think that the authors fail to point some important implications of this model, by not considering a full model of the system, which includes also production and not only uptake.

Here are my main concerns:

1. Pump-probe systems lead to a density independent (but ratiometric) signaling if one considers a full model, irrespective of *phrA* intracellular stability.

If we assume that the extracellular molecules are produced by a fraction f of the population with a production rate p and rapidly degraded by the cells, then the full equation for the extracellular

concentration of phrA is:

$$dCes/dt = Nc/Ve \times p \times f - Nc/Ve \times vmx \times Ces / (K + Ces)$$

If we assume a constant number of cells and if $p < vmx$ then this would lead to a steady state

level of signal which is independent of the cell concentration but depends on the frequency of

producers:

$$Ces(st) = K \times p \times f / (vmx - p \times f)$$

This fits well with a ratiometric model.

The time it will take to reach this steady-state depends on the uptake time scale and therefore on

Nc/Ve , but for concentrations of 10⁸ cells/ml used in this work, the uptake time-scale is <5 minutes.

If there is no degradation of intracellular phrA, then the accumulation implies that intracellular phrA levels will be: $(t) \sim p \times f \times t$

The concentration will increase linearly with time. If PhrA is intracellularly degraded with a timescale

τ_{in} then the internal concentration would reach a steady-state: $C_{int}(st) \sim p \times f \times \tau_{in}$

The ratiometric property is therefore independent of the fact that PhrA is stable in the cell, but

depends only on the fast uptake of PhrA from the medium.

We thank the referee for this excellent suggestion. In general, pump-probe systems could not only generate different functions, but also perform the same function in different ways. We agree that one could introduce the different control functions of pump-probe networks, including quorum sensing, ratio-sensing and chronometric control more intuitively by explicitly modeling signal production. We have therefore extended the original model to consider both production and population growth (see response to C4) and use this model to demonstrate that pump-probe networks could, in principle, execute diverse control functions by varying parameters and operating conditions. The revised model is described in detail in the Supplementary Material and the corresponding simulations are shown in Fig. S1. We also recast the presentation of Results for the different control functions (Lines 145ff), aiming at a less mathematical description, following the suggestions of Referee 1.

2. Stability of PhrA is expected to overwhelm ratiometric measurement. I am not aware of reports on phrA extracellular levels during approach to sporulation, but Grossman and Lazzazera reported phrC levels of ~1 micromolar and Erez et al measured ~50-100nM for the aimP RNPP phage communication system. If one assumes that steady-state extracellular PhrA concentration is 50 nanomolar, then this implies (given the estimated uptake parameters) that $p \sim$

vmx^3

$\sim 6 \times$

10⁴ molecules

min

~ 60

μM

min

.

Under these conditions, the intracellular concentration of phrA would accumulate to inhibitory levels within ~40 seconds and would reach >mM levels in an hour. These **levels are huge compared to the paper's predicted EC50 of Phr on Rap-Spo0F~P** binding (38 μM). Now, if the active population is just 10%, Phr would still accumulate to sufficient levels in ~5 minutes. If only 1% of the population produces Phr, it would still accumulate to sufficient levels to dissociate Rap-Spo0F~P in 50 minutes. Are these numbers reasonable given the long time-scale of the approach to sporulation? I am not

sure. One caveat to the above analysis is that Opp may be occupied by competing peptides during the approach to sporulation, which will reduce its effective import rate. Most of the competing peptides should have been eaten away by the cells by the time sporulation is initiated, though.

The referee is concerned that a ratiometric measurement may not be possible given the parameters inferred in this work, because the pathway could oversaturate. To address this concern, we provide evidence that the RapA-Spo0F pathway will not oversaturate with PhrA.

We agree that the extracellular concentration is a key variable to place our data in their proper context. With the help of a sensitized bioassay we infer a PhrA concentration of 0.4nM under our experimental conditions (Fig. S7). This is lower than estimates from other systems used in the referee's calculation. With the help of the extracellular signal concentration, we estimate the number of signaling molecules that are taken up from the environment (assuming steady-state conditions) by each cell during the signal integration time (timescale set by growth and dilution) and find that $N = 2.7 \cdot 10^4$. This is close to the dose that is required to achieve a half-maximal response, $D_{50} = 2.4 \cdot 10^4$ in our experiments. This calculation suggests that intracellular signal concentrations in wt cells will accumulate roughly to EC_{50} levels under our conditions. We have added the corresponding analysis and results to the Results section (Line 371f).

C9. I expect the author to do the following given the above remarks:

1. Theory: Introduce and analyze a full model of the system – including both signal production, uptake and intracellular accumulation/degradation as well as additional factors such as competition from other peptides. They should discuss the possible implications of the full model for the ratiometric function and the quorum-sensing function. Given their estimated parameters, the impact of intracellular Phr accumulation to levels which far exceed the EC_{50} levels should be discussed in the context of its effect on threshold cell densities and its implication for frequency measurements.

We agree that a suitably revised model is very helpful to demonstrate that pump-probe networks could carry out various functions in a more intuitive manner (see response to comment C8-1, Fig. S1, Text Lines 145ff). Furthermore, work during the revision has shown that effects from dilution due to cell growth should be included in order to characterize signal processing by the PhrA-signaling system on longer time scales (Fig. 3C, see response to comment C4). Based on this revised analysis, intracellular PhrA levels will not exceed the EC_{50} levels by far (see response to comment C8-2). Instead, the values are consistent with the ability of the system to carry out a ratio-sensing function (Line 371f). Thus – while one could speculate on what might happen in the hypothetical case of very high intracellular signal concentrations – we feel that such a speculative discussion would not add much to further improving the manuscript. We would like to thank the referee once again for excellent comments and suggestions, which have helped to improve the manuscript very much.

2. Experiments: Measure the effective concentration of phrA during approach to sporulation to be able to better estimate the relevant parameter regime. They can do that by using conditioned medium from sporulation in their FRET assay in a similar manner to what they have done in the paper to measure PhrA depletion from the conditioned medium.

We agree that measurements of extracellular PhrA levels under physiological conditions are required in order to argue convincingly that that ratiometric population sensing is feasible given the inferred parameters from the model (see response to C8). As suggested, we have determined the PhrA levels in supernatants from wt cells under the experimental conditions used for probing signal processing with the FRET reporter (i.e. cultivation in S7 media to an $OD_{600} = 1.6$). Using a sensitized bioassay (described in Lines 599f) we estimate that PhrA is

present at sub-nM concentration, $C_e \sim 0.4$ nM (Fig. S7). As mentioned above, the analysis of the system under sporulation conditions is beyond the scope of this manuscript, and must be addressed in future work.

Response to Referee 3

This is a well-written manuscript proposing that pump-probe networks can be modulated to perform different sensory tasks and mediate different regulatory functions, including quorum sensing, chronometric and ratiometric control. Authors describe the characteristics of the signal transduction system of prototypical PhrA-RapA system in *Bacillus subtilis*, using FRET to quantify extra- to intracellular signal conversion. FRET results and methodology are fundamental for the understanding as well as significance of this paper.

Therefore, this paper would be significantly improved by addressing the following issues relating to the FRET experiments:

We thank referee 3 for the assessment of our work and for the constructive comments, especially with respect to the FRET results and the methodology. In the revised version, we have taken care to improve our presentation in order to clarify misleading points, and have performed additional control experiments to address the specific issues raised, as outlined below. The new data corroborates our previous findings. We would like to point out upfront that we have measured the FRET efficiency based on acceptor photobleaching experiments on populations of bacteria, and we apologize for not making this sufficiently clear in the original manuscript.

C1. FRET experiments should include a single-donor as well as a single acceptor experiment to demonstrate lack of spectra-bleedthrough or FRET contamination. These controls would also indicate the background level of the FRET experiments indicating **the meaning of “no FRET level”**.

We agree that appropriate controls are required to indicate the background level of FRET in acceptor photobleaching experiments. We used a negative control based on expressing non-interacting free YFP and CFP in the same experimental design as used for the FRET reporter. This negative control reports any possible FRET contaminations due to bleedthrough or photoconversion but also non-specific FRET induced by molecular crowding. The percentage change in fluorescence in the donor channel was $(0.04 \pm 0.3) \%$ (see Fig. S3B, second bar from the top, and Fig. 2C), while we scored $>0.6 \%$ increase in donor fluorescence as significant FRET, as previously in Kentner & Sourjik, 2009. To improve clarity, we duplicate our data on the FRET negative control in Fig. S3B and in Fig.2C in the Results Section. We now have further included a donor-only control, as suggested by the referee, which is again clearly negative (Fig. S3B). Since only CFP fluorescence is measured in the acceptor photobleaching FRET, including acceptor-only sample would not be informative. Together, this data confirms that, under our experimental conditions, the percentage change in CFP fluorescence provides an reliable measure of the FRET efficiency. To improve clarity, we revised the Results section Line 204f and refer to Fig. 3SB for details.

C2. FRET microscopy is an imaging experiment. Images should be shown in supplementary material, at least for one relevant experiment.

We apologize for not describing it more clearly, but in our application of acceptor photobleaching FRET we measured the integral fluorescence of populations of hundreds of cells using photomultiplier tube, in a setup analogous to the one described in more detail by Kentner and Sourjik, 2009. So no fluorescence images have been acquired. To improve clarity, we explicitly mention this important detail when introducing our methodology in the Results Section, Line 202. Moreover, we modified Fig. S3A to replace the term “detector” by “PMT”. Finally, in response to comment C5 of the referee (see below), we have performed an imaging experiment to determine E-FRET. Representative images are included in the Supplementary Material Fig. S8A and B.

C3. Role of acceptor concentration levels on FRET experiments need to be address to indicate the relevance of molecular crowding effect on FRET results.

We agree that molecular crowding might be potentially an issue, and the experiment proposed by the referee may be useful for determining optimal acceptor levels in studies involving high-affinity complexes. In that context, FRET increases with increasing acceptor concentration and plateaus as the level of acceptor exceeds the donor concentration. In contrast, in a crowded environment with non-specific interactions FRET always increases. A plateau in FRET is thus an indicator for a specific interaction. However, many signaling complexes are governed by “quinary interactions” which often results in weaker binding (Sukenik et al., PNAS 2017). Thus, under physiological conditions, complex formation is typically not saturated. Therefore, when studying such complexes by FRET, it is technically difficult to saturate the donor by (over)expressing the acceptor. Moreover, the specific interaction regime and the crowding regime may not be separated by a FRET plateau. Hence, in this case, the relevance of crowding must be addressed by other means.

Our FRET negative control (Fig. S3B) was designed to test for non-specific FRET by monitoring non-interacting CFP and YFP expressed under same experimental conditions as those used for the FRET reporter (Fig. S3B, second bar from the top) and thus at similar levels as the FRET pair. Little, if any FRET ($\text{FRET} = 0.04 \pm 0.3$) is observed in this control. This indicates that the donor and acceptor concentrations are too low to give rise to a crowding-dependent non-specific FRET signal. To improve clarity, molecular crowding is explicitly mentioned in the Results section when discussing the FRET negative control (line 208).

Shahar Sukenik, Pin Ren, Martin Gruebele

Detection of weak protein complexes in live cells

Proceedings of the National Academy of Sciences Jun 2017, 114 (26) 6776-6781

<https://www.pnas.org/content/114/26/6776>

C4. What parameter is being shown as FRET[-]? What does FRET[-] represent? FRET efficiency or some apparent relative FRET parameter that should then be clearly defined. How are these FRET values calculated? FRET efficiency, E%, or FRET ratios? We measure the FRET efficiency as the percentage increase in CFP fluorescence after acceptor photobleaching, i.e. $\text{FRET}[\%] = (\text{CFP}_{\text{post}} - \text{CFP}_{\text{pre}}) / \text{CFP}_{\text{post}} \times 100\%$, as detailed in the Methods section “Quantification of FRET”. To clarify the point, we assign an equation number Eq.(1) to the definition, and refer to that number when introducing acceptor photobleaching (Line 174f). We agree that the FRET [-] notation was confusing and have changed it to FRET [%] in all figures.

C5. Alternative FRET measurements using lifetime imaging or intensity based microscopy should be used to validate acceptor photobleaching.

We agree. We have used E-FRET imaging to corroborate our results from fluorometry using acceptor photobleaching. We determined E-FRET for both unstimulated and stimulated cells following the approach introduced by Zal and Gascoigne (2004). We found very good agreement between E-FRET and the FRET efficiency from acceptor photo-bleaching measurements. E-FRET imaging experiments are described in the Supplementary Methods and the corresponding data is included in Fig. S8, respectively.

Zal, T and Gascoigne, N. R. (2004), ‘Photobleaching-corrected FRET efficiency imaging of live cells.’, Biophysical Journal, 86(6):3923-39 [https://www.cell.com/biophysj/fulltext/S0006-3495\(04\)74433-1](https://www.cell.com/biophysj/fulltext/S0006-3495(04)74433-1)

C6. The FRET reporter is based on inter-molecular FRET so molecular crowding has to be addressed and excluded.

We agree. As explained in C3, molecular crowding can be excluded.

C7. Also bimolecular reporter should not be used to describe these FRET experiments since it may suggest fusion protein constructs.

We agree. We have removed the term bimolecular reporter from the manuscript.

C8. Authors describe FRET results as percent increase which do not allow for statistical evaluation and can be misleading. FRET results should be present more directly as fold increases/decreases and as much as possible absolute FRET efficiency numbers should be included allowing for readers to evaluate results.

Unfortunately, we find this comment difficult to interpret. It appears to imply that the referee believes that the FRET data presented were normalized in some way. This is not the case. All our FRET data reports the absolute FRET efficiency as measured by acceptor photobleaching by calculating the percentage increase in fluorescence in the donor channel relative to the value prior to bleaching (see response to comment C4).

As pointed out by the referee, a key advantage of using acceptor photobleaching experiments is that this method provides an absolute measure of FRET. Thus, all FRET results are directly comparable and can be statistically evaluated. Indeed, it was the ability to compare FRET results across different experiments that enabled us to derive a quantitative description of the FRET response using the pump-probe model. As noted in our reply to comment C5, thus obtained values of FRET efficiency are very similar to our new results using E-FRET imaging. To improve clarity, we explicitly state in the Results section that acceptor photobleaching experiments provide an absolute measure of the FRET efficiency and thus facilitate direct comparisons of data across experiments (Lines 175f).

C9. Statistical analysis should be provided and clearly described. It is not clear that actual statistical analysis has been performed to validate significance of results.

In most experiments (Figs. 3, 4 and 5), we determine a functional relationship between the FRET efficiency and one external variable, e.g. time, signal concentration, concentration of competitor peptides etc. The functional dependence of FRET on the relevant variables was evaluated with the help of the pump-probe model. We arrive at a parameterized model by data fitting, which we use to predict each functional relationship (lines in the figures). Furthermore, we perform bootstrapping of our experimental data to derive the 95% confidence intervals for each predicted functional relationship. This procedure explained in the subsection of the Materials and Methods that is now named Model Fitting and Statistical Analysis .

When applicable we performed a t-test or one-way ANOVA to test for significance of observed differences. We have now included information on the level of significance in the respective figures (Fig. 2C, Fig. 4C, Fig. S4, Fig. S5C, Fig. S7 and Fig. S8) and provide information on the statistical analysis/tests performed in a separate section Statistical Analysis of the Materials and Methods.

REVIEWERS' COMMENTS:

Reviewer #2 (Remarks to the Author):

I have carefully read the rebuttal and new version of the manuscript.

In general, I feel that all my major concerns have been answered:

1. Phosphorylation is not happening under the conditions studied and therefore the additional layer of complexity it adds does not affect the results. I agree with this statement, but I still think the authors should briefly mention in the discussion what will be the additional layer of complexity when the system is studied in its natural context. Specifically, previous observation that the affinity of Rap to Spo0F~P is considerably higher than to Spo0F should be discussed.
2. Degradation occurs but at a low rate. I generally accept this observation.
3. External signal level is very low, solving the design problem of rapid Rap saturation. I accept the authors measurements that show that under their experimental condition PhrA levels are low. Going, again, to the natural function of the system, the authors should mention that PhrA expression is controlled by the phosphorelay, probably leading to higher levels of the peptide under sporulation conditions.

A minor note – there are some wrong references in the text to some of the figures.

I think that the flow of the manuscript is improved now and the additional changes clarified a variety of other issues. I therefore accept the manuscript for publication with minor corrections. I do not see a reason to review it again.

Reviewer #3 (Remarks to the Author):

The authors have addressed in detail all reviewer's comments

Response to Referees

We would like to thank the Reviewers for dealing with our revised manuscript and for the comments provided. The revised manuscript has been modified in accordance with the comments of Referee 2.

The changes to the manuscript are summarized below:

- We added a statement to the Introduction that the transcription of rapA-phrA-operon is highly regulated and active under both non-sporulating and sporulating conditions. We explicitly state that we study the system under non-sporulating – yet still physiologically relevant – conditions and we define these conditions in more detail in the Results section.
- We discuss that our experiments likely probed the interaction with unphosphorylated Spo0F. We refer to the work by Ishikawa et al. that suggests that phosphorylation changes the interaction with RapA and point out that this could affect FRET.
- We discuss that the inferred parameters for the PhrA-RapA network may change at the transition from non-sporulating to sporulating conditions, since all signaling components (and their interactions) are themselves regulated by a complex network. Such a change of network parameters could have an effect on network function, according to the model.

These changes have been highlighted in the revised manuscript.

Please find our point-by-point responses to the Referees' comments below. We hope that the revised manuscript is now acceptable for publication in Nature Communications.

Reply to the comments of Referee 2

I have carefully read the rebuttal and new version of the manuscript. In general, I feel that all my major concerns have been answered:

We thank the Referee for the assessing the revised manuscript and for the comments provided.

C1. Phosphorylation is not happening under the conditions studied and therefore the additional layer of complexity it adds does not affect the results. I agree with this statement, but I still think the authors should briefly mention in the discussion what will be the additional layer of complexity when the system is studied in its natural context. Specifically, previous observation that the affinity of Rap to Spo0F~P is considerably higher than to Spo0F should be discussed.

The Referee agrees that our FRET reporter measurements are specific to PhrA as - under our experimental conditions - knockouts of phosphorelay genes do not affect FRET. However, the Referee's comment seems to imply that non-sporulating conditions are artificial conditions for studies of PhrA signaling. This is an assertion that we do not share. It is important to point out that the *rapA-phrA* operon is induced under non-sporulating conditions (Comella et al. 2005, Griffith et al. 2008, Wolf et al. 2016) and PhrA is produced by wild-type populations (Supplementary Figure 7). Thus, the experimental conditions studied here could be physiologically relevant.

Moreover, strictly speaking, from our data one cannot conclude that "phosphorylation is not happening" under our conditions (although this is likely, given that cells do not sporulate). In principle, our data does not rule out that phosphorylation of Spo0F has no effect on FRET. If Spo0F~P had a higher affinity to RapA and the complex formed was otherwise identical, FRET would increase. However, if complexes formed by Spo0F and Spo0F~P showed different FRET, the cellular FRET signal could, in theory, also decrease upon phosphorylation. Clearly, experiments are required to test whether and how phosphorylation of Spo0F affects FRET.

In the revised manuscript we better clarify the relationship between non-sporulating and sporulating conditions. We added a statement to the Introduction that the transcription of *rapA-phrA*-operon is highly regulated and active under both non-sporulating and sporulating conditions. We explicitly state that we study the system under non-sporulating – yet still physiologically relevant – conditions and we define these conditions in more detail in the Results section.

Following the suggestion of the Referee, in the Discussion we state that our experiments likely probe the interaction with unphosphorylated Spo0F. We refer to the work by Ishikawa et al. that suggests that phosphorylation changes the interaction with RapA and point out that this could affect FRET.

C2. Degradation occurs but at a low rate. I generally accept this observation.

C3. External signal level is very low, solving the design problem of rapid Rap saturation. I accept the authors measurements that show that under their experimental condition PhrA levels are low. Going, again, to the natural function of the system, the

authors should mention that PhrA expression is controlled by the phosphorelay, probably leading to higher levels of the peptide under sporulation conditions.

The Referee accepts that PhrA levels in culture supernatants are low under our experimental (non-sporulating) conditions. The Referee expects that PhrA levels will be higher under sporulating conditions as PhrA expression is regulated by the sporulation phosphorelay via Spo0A (Molle et al. 2003, Fujita et al. 2005).

We believe that the Referee's argument is an over-simplification for the following reasons:

1) Spo0A~P is not the only input to the rapA-phrA operon, instead the main input required for activation of the promoter is the transcription factor ComA (Comella et al. 2005, Griffith et al. 2008, Wolf et al., 2016). The ComA-pathway is activated under non-sporulating conditions (Griffith et al. 2008, Wolf et al., 2016).

2) Low Spo0A~P levels seem to enhance transcription. However, strong activation of Spo0A~P (as under sporulation conditions) represses transcription of the operon (Molle et al. 2003, Fujita et al. 2005). Moreover, transcription becomes heterogeneous across the population (Bischofs et al. 2009, Mutlu et al. 2018) as a subpopulation of cells enters sporulation and downregulates the operon and thus, presumably the production of PhrA.

3) Production of mature PhrA is additionally regulated at the post-transcriptional level (Lannigan-Gerdes et al. 2007).

4) PhrA levels in the supernatant depend on the production rate but also other parameters, e.g. uptake (as shown by our model). The values for these parameters may change as conditions change, since all signaling components are themselves regulated by a complex network.

We agree with the Referee that under sporulation conditions, PhrA levels and network parameters may be different from the values we determined under non-sporulating conditions. However, the available data does not justify a discussion about "likely results".

In the Discussion we therefore state that at the transition from non-sporulating to sporulating conditions, the inferred parameters for the PhrA-RapA network may change, since all signaling components (and their interactions) are themselves regulated by a complex network. Such a change of network parameters could have an effect on network function, according to the model.

C. A minor note – there are some wrong references in the text to some of the figures. Thanks. We have checked and corrected all references.

I think that the flow of the manuscript is improved now and the additional changes clarified a variety of other issues. I therefore accept the manuscript for publication with minor corrections. I do not see a reason to review it again.

We thank Referee 2 for the interest in our work and the constructive review process.

Reviewer #3 (Remarks to the Author):

The authors have addressed in detail all reviewer's comments

We thank Referee 3 for the interest in our work and the constructive review process.